# Weakly Supervised Anomaly Detection via Dual-Tailed Kernel

**Walid Durani** [1]   **Tobias Nitzl** [2]   **Claudia Plant** [3]   **Christian Böhm** [4]

## Abstract

Detecting anomalies with limited supervision is challenging due to the scarcity of labeled anomalies, which often fail to capture the diversity of abnormal behaviors. We propose Weakly Supervised Anomaly Detection via Dual-Tailed Kernel (WSAD-DT), a novel framework that learns robust latent representations to distinctly separate anomalies from normal samples under weak supervision. WSAD-DT introduces two centroids—one for normal samples and one for anomalies—and leverages a **dual-tailed kernel scheme**: a light-tailed kernel to compactly model in-class points and a heavy-tailed kernel to maintain a wider margin against out-of-class instances. To preserve intra-class diversity, WSAD-DT incorporates kernel-based regularization, encouraging richer representations within each class. Furthermore, we devise an ensemble strategy that partitions unlabeled data into diverse subsets, while sharing the limited labeled anomalies among these partitions to maximize their impact. Empirically, WSAD-DT achieves state-of-the-art performance on several challenging anomaly detection benchmarks, outperforming leading ensemble-based methods such as XGBOD.

## 1. Introduction

Anomaly detection identifies data instances that deviate substantially from normal patterns (Agrawal & Agrawal, 2015), with critical applications in credit risk analysis (John & Naaz, 2019), network intrusion detection (Tao et al., 2018), and medical diagnostics (Abuzaid, 2020). Fully unsupervised methods often yield high false positives, while fully su-

pervised approaches require many labeled anomalies, which is typically infeasible (Han et al., 2022). Consequently, *weakly supervised* anomaly detection has emerged, leveraging a small labeled anomaly set alongside predominantly normal unlabeled data (Pang et al., 2023). Two notable examples are *DeepSAD* (Ruff et al., 2019), which extends *DeepSVDD* (Ruff et al., 2018) by pushing labeled anomalies away from a single "normal" center, and *DevNet* (Pang et al., 2019), modeling anomalies as the extreme tail of a univariate distribution. Both can handle sparse labels but rely on a single center or tail, risking performance degradation when anomalies exhibit significant heterogeneity or cause collapsed embeddings (Goyal et al., 2020). Motivated by these challenges and guided by classical margin-based theory, which advocates maintaining a tight radius around in-class samples while enforcing a wide margin against out-of-class points to reduce model complexity and enhance generalization (Cristianini & Shawe-Taylor, 2000; Bartlett & Mendelson, 2002; Tax & Duin, 2004)—we introduce Weakly Supervised Anomaly Detection via Dual-Tailed Kernel (*WSAD-DT*). Our method is carefully tailored to the nuances of weakly supervised anomaly detection, where labeled anomalies are not only scarce but may also underrepresent the true diversity of anomalous behaviors. The core idea of WSAD-DT is to map input data into a latent representation that effectively distinguishes normal from anomalous samples. To achieve this, our approach employs dynamic similarity measures based on heavy-tailed and light-tailed kernels (Schölkopf & Smola, 2002) (Fig. 1), which are tailored to the distinct characteristics of anomalies and normal samples. Specifically, WSAD-DT leverages a light-tailed kernel to tighten the representation around each class center by rapidly reducing similarity with increasing distance. Concurrently, a heavy-tailed kernel ensures slower similarity decay, regulating broader dispersion farther away—balancing compactness for in-class points with greater margins against out-of-class samples. This dual-tailed kernel mechanism effectively balances compactness near each center while allowing broader margins for out-of-class patterns, thereby enabling clear separation between normal and anomalous instances. Fig. 1 illustrates the decay behaviors of light and heavy-tailed kernels, highlighting their suitability for modeling normal and anomalous data, respectively. Additionally, WSAD-DT incorporates kernel-based regularization to promote intra-class diversity and avoid over-concentration,

---

[1]LMU Munich, Munich Center for Machine Learning (MCML), Munich, Germany [2]LMU Munich, Munich, Germany [3]Faculty of Computer Science, ds:UniVie, University of Vienna, Vienna, Austria [4]Faculty of Computer Science, University of Vienna, Vienna, Austria. Correspondence to: Walid Durani <durani@dbs.ifi.lmu.de>.

*Proceedings of the 42$^{nd}$ International Conference on Machine Learning*, Vancouver, Canada. PMLR 267, 2025. Copyright 2025 by the author(s).

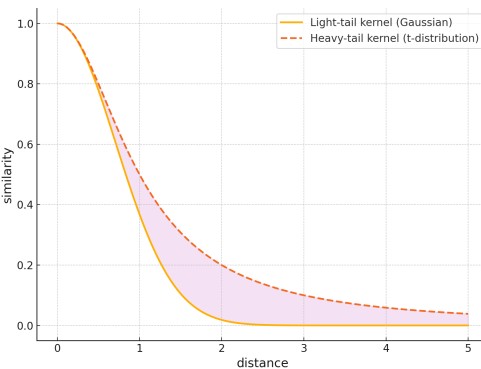

*Figure 1.* Comparison of two distance-based similarity functions—light-tailed and heavy-tailed—as distance increases. The light-tailed kernel's similarity rapidly falls off at moderate distances. In contrast, the heavy-tailed decays more gradually, preserving moderate similarity for points farther away.

ensuring a robust latent space representation. To address the variability and imbalance inherent in weakly supervised settings, WSAD-DT also introduces an effective ensemble learning mechanism. This ensemble approach leverages diverse subsets of the data while sharing labeled anomalies across all subsets, which enhances robustness and surpasses traditional ensemble methods like XGBOD (Zhao & Hryniewicki, 2018) in anomaly detection tasks.

Our key contributions are as follows:

- **Dynamic Similarity Measures**: We introduce a novel loss function that incorporates both light-tailed and heavy-tailed kernels, ensuring compact representation around each class center while allowing broader dispersion for out-of-class samples. This dual-tailed kernel design captures typical data tightly yet accommodates anomalous deviations more flexibly, thereby improving separation under limited supervision.

- **Kernel-Based Regularization**: To prevent degenerate "all-points-collapse" solutions, we incorporate a kernel-based regularization term that promotes intra-class diversity. This term ensures that each class retains its inherent variability in the latent space, thereby enhancing robustness and generalization.

- **Ensemble-Based Learning with Subset Splitting**: An effective ensemble scheme divides unlabeled data into multiple subsets, while the limited labeled anomalies are shared across all partitions. This design yields diverse yet anomaly-aware models, and aggregated decisions outperform strong ensemble baselines such as XGBOD in anomaly detection tasks.

## 2. Related Work

Detecting anomalies under weak supervision—where a small subset of anomalies is labeled while the majority of data remain unlabeled—has attracted growing interest in recent years (Pang et al., 2023).

DeepSAD (Ruff et al., 2019) extends DeepSVDD (Ruff et al., 2018) by introducing an autoencoder-based latent representation. In this representation, a single learned center pulls normal samples closer and pushes anomalies farther away; a hypersphere is then fitted to primarily capture normal embeddings. To address the collapse issue in DeepSVDD, DROCC (Goyal et al., 2020) introduces adversarial perturbations around normal points to learn robust decision boundaries. Subsequently, DROCC-LF (Goyal et al., 2020) incorporates a small number of labeled anomalies alongside local feature representations, thus maintaining adaptable boundaries even in high-dimensional spaces. DevNet (Pang et al., 2019) employs a Gaussian prior in the latent space and a *deviation loss* that pulls normal samples toward the distribution center while pushing anomalies outward. This focus on relative deviations yields competitive performance. FeaWAD (Zhou et al., 2021) builds on the same principle, fusing the deviation loss with autoencoders for more robust representations. PReNet (Pang et al., 2023) tackles the anomaly detection problem from a meta-learning viewpoint, combining pairwise relations with a self-supervised distance metric to more effectively isolate anomalies. A separate line of research leverages Generative Adversarial Networks (GANs) (Goodfellow et al., 2014). For example, GANomaly (Akcay et al., 2019) learns a latent space in which the generator–discriminator pair accentuates differences between reconstructions of normal samples and anomalies. While GAN-based models can capture rich data distributions, they often require careful tuning for stable adversarial training. RoSAS (Xu et al., 2023b) further refines semi-supervised AD by introducing a contamination-resilient, continuous supervision mechanism that uses mass interpolation to produce smoothly varying anomaly scores, enhancing robustness to label noise. Beyond deep end-to-end solutions, hybrid approaches remain popular. XG-BOD (Zhao & Hryniewicki, 2018), for instance, combines a gradient-boosting ensemble with unsupervised anomaly scores, integrating weak labels to sharpen decision boundaries. **Single- vs. Multi-Kernel Methods.** Classical kernel-based methods such as One-Class SVM (Schölkopf et al., 2001) rely on a *single* kernel for all data. Meanwhile, various multi-view or multi-kernel approaches blend multiple kernels—often via linear combinations—to capture richer similarities (Gönen & Alpaydın, 2011). For example, Dual-Regularized Multi-View Outlier Detection *DMOD* (Zhao & Fu, 2015) factors multi-view data into cluster indicators and sample-specific errors, thereby modeling both *cross-view inconsistencies* and *universal anomalies*. By contrast, our

work assigns *two distinct* kernels—light-tailed for in-class points and heavy-tailed for out-of-class points—thus explicitly enforcing separate tail behaviors for each class rather than learning a single "blended" kernel for the entire dataset.

## 3. The Proposed Approach

### 3.1. Problem Setting

Let $\mathcal{X} \subset \mathbb{R}^D$ be the input feature space, and consider a dataset $\mathcal{D} \subset \mathcal{X} \times \{0, 1\}$. In weakly supervised anomaly detection, the dataset $\mathcal{D}$ is divided into an unlabeled set $\mathcal{D}_U = \{(x_1, 0), \ldots, (x_N, 0)\}$, which is largely normal but may contain a few anomalies, and a labeled set $\mathcal{D}_L = \{(x_{N+1}, 1), \ldots, (x_{N+K}, 1)\}$, comprising a small number of confirmed anomalies. Since $K \ll N$, these labeled anomalies provide only sparse supervision and often span a narrow range of anomaly types (Pang et al., 2023). The objective is to learn a scoring function $\phi : \mathcal{X} \to \mathbb{R}$ that ranks true anomalies (labeled or unseen) higher than normal samples. This goal poses two key difficulties: (i) learning a robust decision boundary from a small set of labeled anomalies, and (ii) generalizing effectively on a large, unlabeled pool that may include unknown anomalies.

### 3.2. Overview of WSAD-DT

To tackle these challenges, this paper introduces *Weakly Supervised Anomaly Detection via Dual-Tailed Kernel (WSAD-DT)*, a framework tailored for scenarios with scarce labeled anomalies and abundant unlabeled data dominated by normal instances. WSAD-DT learns a feature representation $f : \mathcal{X} \to \mathcal{Z}$, where $\mathcal{X} \subseteq \mathbb{R}^D$ and $\mathcal{Z} \subseteq \mathbb{R}^d$, that separates normal from anomalous instances under limited supervision and data imbalance. The mapping $f$ is a **feature representation learner**, instantiated as a neural network $f(\cdot; \Theta_f)$ with parameters $\Theta_f = \{W_1, \ldots, W_H\}$, where $H \in \mathbb{N}$ denotes the number of hidden layers. WSAD-DT employs two distinct latent centers—one representing cohesive normal patterns and another capturing dispersed anomalies—to establish a clear boundary between normality and anomalies. A dual-tailed kernel approach then sharpens this divide by keeping each class tightly localized around its matching center while pushing out-of-class samples away from the opposing center. However, merely driving data toward these centers can trigger degenerate *collapses*, where all points of a class map to a single coordinate. To avert this, WSAD-DT includes a *diversity term* that preserves intra-class variability. Additionally, an ensemble strategy further enhances stability and generalization by distributing the limited anomaly labels across multiple partitions of unlabeled data. Finally, WSAD-DT translates the learned embeddings ($\mathcal{Z}$) into anomaly scores $\phi : \mathcal{Z} \to \mathbb{R}$ by contrasting each embedding's similarity to the normal center against its similarity to the anomaly center.

## 4. Two Centers based Separation

In WSAD-DT, two dedicated centers $c_0$ (for normal data) and $c_1$ (for anomalies) are placed in the latent space to better capture the distinct characteristics of each class, particularly when labeled anomalies are scarce. In the weakly unsupervised setting, we assume that the unlabeled dataset $\mathcal{D}_U$ is predominantly composed of normal instances (Pang et al., 2023), although it may contain some anomalies. Nevertheless, this assumption provides a solid foundation for modeling normal behavior. In Appendix M, we further analyze how WSAD-DT performs under varying levels of contamination in the unlabeled data. A neural network $f(\cdot; \Theta_f)$ projects each input $x_i$ to a latent embedding $z_i$, and the centers are initialized by averaging embeddings from $\mathcal{D}_U$ (predominantly normal) and $\mathcal{D}_L$ (confirmed anomalies), as in $c_0 = \frac{1}{|\mathcal{D}_U|} \sum_{(x_i, y_i) \in \mathcal{D}_U} f_\theta(x_i)$ and

$c_1 = \frac{1}{|\mathcal{D}_L|} \sum_{(x_i, y_i) \in \mathcal{D}_L} f_\theta(x_i)$, ensuring $c_0 \neq c_1$. By aligning each class with its dedicated center, normal samples cluster around $c_0$ and anomalies are pulled toward $c_1$, yielding a sharper distinction between their intrinsic characteristics. This two-center design effectively decouples normal and anomalous representations in the latent space, minimizing overlap and improving separation.

**Remark Single Center for Normality.** Although real data can exhibit multiple modes of normal behavior, we adopt a single center $c_0$ to represent normal samples because allowing multiple centers would introduce additional complexity and the risk of overfitting or unstable partitions in weakly supervised scenarios. This choice aligns with classic one-class frameworks (e.g., DeepSVDD and DeepSAD) that effectively use a single hypersphere or reference point for normal data and still achieve strong performance. Empirically, our experiments (Tables 1, 5) demonstrate that even a single center, combined with the dual-tailed kernel and a suitable diversity term, can robustly capture local variations while maintaining enough variance to avoid degeneracy. Thus, one center often strikes the right balance between simplicity, robustness, and generalization, especially under limited anomaly labels.

## 5. Dual-Tailed Kernel: Motivation and Formal Definition

Margin-based theory (Cristianini & Shawe-Taylor, 2000; Bartlett & Mendelson, 2002; Tax & Duin, 2004) emphasizes the importance of forming a compact in-class region while maintaining a broad margin for out-of-class points. Motivated by these insights, the main objective of WSAD-DT is to achieve high in-class compactness and a clear boundary

for anomalous samples, thereby yielding robust representations even under limited anomaly labels. To this end, we propose a dual-tailed kernel separation loss that adaptively switches between two complementary similarity functions according to each sample's alignment with its assigned class center. Concretely, we first define a distance metric to quantify how far each sample lies from its designated class center. We then transform these distances into similarity scores through two specialized kernels: one that reinforces a tighter distribution within each class, and another that preserves a wider margin for out-of-class points. This dual perspective ensures both strong in-class compactness and a robust boundary for anomalous samples, ultimately producing discriminative embeddings even with sparse anomaly supervision.

## 5.1. Distance Metric

For each sample $(x_i, y_i)$, let $f_\theta$ be the neural network mapping $x_i \in \mathbb{R}^D$ into the latent space. The distance between $f_\theta(x_i)$ and $c_k$ ($k \in \{0, 1\}$) is defined as:

$$d_{i,k} = \left\| f_\theta(x_i) - c_k \right\| = \sqrt{\sum_{j=1}^{d} \left( f_\theta(x_i)_j - c_{k,j} \right)^2}. \quad (1)$$

where "$\| \cdot \|$" denotes the standard Euclidean ($L^2$) norm. Because $y_i \in \{0, 1\}$ indicates whether $x_i$ is normal ($y_i = 0$) or anomalous ($y_i = 1$), we assign:

$$d_{\text{in}} = \begin{cases} d_{i,0}, & \text{if } y_i = 0, \\ d_{i,1}, & \text{if } y_i = 1, \end{cases} \quad d_{\text{out}} = \begin{cases} d_{i,1}, & \text{if } y_i = 0, \\ d_{i,0}, & \text{if } y_i = 1. \end{cases} \quad (2)$$

Hence, $d_{\text{in}}$ measures how close the sample is to its own class center, while $d_{\text{out}}$ measures its distance to the *opposite* (out-of-class) center (i.e. $c_1$ for normal data, $c_0$ for anomalous). *We aim to decrease $d_{\text{in}}$* (pulling each sample to its own center) *and increase $d_{\text{out}}$* (pushing it away from the out-of-class center), thus causing normal data to cluster near $c_0$ and anomalies near $c_1$.

## 5.2. Similarity-Based Kernels

To achieve compact in-class representation while maintaining a broader margin for out-of-class points, we convert distances in the latent space into a *similarity* measure using carefully chosen kernels. These kernels regulate how fast similarity decays with distance: To capture how data behaves in relation to these centers, we employ *distance-based kernels*. Let

$$\mathcal{K} : \mathbb{R}_{\geq 0} \rightarrow \mathbb{R}_{\geq 0}$$

be a function that maps a nonnegative distance ($d$) to a nonnegative similarity score, satisfying $\mathcal{K}(0) = 1$ (maximum similarity) and $\mathcal{K}(d) \to 0$ as $d \to \infty$. We use 'kernel' informally here: $\mathcal{K}$ need not be positive-semidefinite, but is

simply a distance-based similarity. In our setting, we use two variants that differ in their tails (light-tailed vs. heavy-tailed), reflecting different asymptotic decay behaviors. By applying these kernels to distances $d_{i,k}$, we obtain a continuous measure of how "close" a sample is to its in-class center or out-of-class center, allowing us to capture both near- and long-range relationships in the latent space.

**Light-tailed kernel (Compact Representations):** We utilize a *light-tailed kernel* designed to enforce a rapid decay in similarity as points move farther from their true center. A kernel $\mathcal{K}_{\text{light}}$ is called *light-tailed* if it decays at least as fast as an exponential function for large $d$. Formally, there exists a constant $c > 0$ such that

$$\lim_{d \to \infty} \frac{\mathcal{K}_{\text{light}}(d)}{e^{-c\,d}} = L, \quad (3)$$

where $0 < L < \infty$. Equivalently, $\mathcal{K}_{\text{light}}(d)$ approaches 0 *faster* than any polynomial rate $d^{-p}$. This rapid decay enforces tight representations: small distances yield high similarity, while similarity plummets quickly as $d$ grows (more detail in Section 6.1 and Appendix H).

**Heavy-Tailed Kernel (Separation Enforcement)** : In a two-class setting, it is crucial that off-center points remain distinctly separated from the wrong class. While a light-tailed kernel excels at pulling well-matched points close to its center, it may fail to emphasize separation once points deviate substantially. In contrast, a heavy-tailed kernel decays more gradually with distance, preserving a moderate similarity for out-of-class points even at larger distances and thereby enforcing a clearer boundary. Formally, we say $\mathcal{K}_{\text{heavy}}$ is **heavy-tailed** if it decays at most polynomially: there exist constants $p > 0$ and $\ell > 0$ such that

$$\lim_{d \to \infty} \left( d^p \, \mathcal{K}_{\text{heavy}}(d) \right) = \ell, \quad (0 < \ell < \infty). \quad (4)$$

This implies $\mathcal{K}_{\text{heavy}}(d)$ remains larger at moderate to large distances than an exponentially decaying function, maintaining a "long tail" that keeps out-of-class points distinguishable. Consequently, mismatched samples retain consistently low (but not vanishing) similarity to the wrong center, reinforcing class separation (more detail in Section 6.1 and Appendix H)).

By further considering the ratio of light and heavy-tailed kernels:

**Lemma 5.1** (Ratio of Light- and Heavy-Tailed Kernels). *Let $\mathcal{K}_{light}$ be light-tailed and $\mathcal{K}_{heavy}$ be heavy-tailed as defined. Then*

$$\lim_{d \to \infty} \frac{\mathcal{K}_{light}(d)}{\mathcal{K}_{heavy}(d)} = 0. \quad (5)$$

*Proof.* By definition, $\mathcal{K}_L(d)$ decays at least exponentially with rate $\alpha > 0$, while $\mathcal{K}_H(d)$ is at most polynomial decay. Hence

$$\frac{\mathcal{K}_{\text{light}}(d)}{\mathcal{K}_{\text{heavy}}(d)} \sim \frac{e^{-\alpha d}}{d^{-\beta}} = d^{\beta}\, e^{-\alpha d} \xrightarrow[d \to \infty]{} 0. \quad (6)$$

$\square$

confirms, that light-tailed kernels rapidly decay to near-zero for distant points, while heavy-tailed kernels diminish more gradually, preserving longer-range distinctions.

This result underpins why one kernel function cannot, at the same distance scale, exhibit both "very small" similarity values (the hallmark of an exponential/light tail) and "moderate" similarity values (the hallmark of a polynomial/heavy tail). If a kernel had to be "light-tailed" for in-class compactness and "heavy-tailed" for out-of-class separation at the same distance regime, it would violate the limit. Hence, Lemma 5.1 is critical: it clarifies that no single kernel can act like both an exponential tail and a polynomial tail simultaneously. .

## 6. Dual-tailed kernel

In the following, we consider two disjoint classes of samples in the feature space $\mathbb{R}^d$: the normal class
$B = \{x_i | (x_i, y_i) \in \mathcal{D}_U\}$ and the anomalous class $\mathcal{A} = \{x_i | (x_i, y_i) \in \mathcal{D}_L\}$ (known labeled anomalies).

### 6.1. Single-tailed kernel vs. Dual-tailed kernel Approaches

In the following, we demonstrate why (1) *light*-tailed kernels excel at in-class compactness, (2) *heavy*-tailed kernels excel at out-of-class separation, and (3) no single kernel can provide both simultaneously. This naturally motivates a *dual-tailed kernel* approach (light-tailed kernel for in-class, heavy-tailed kernel for out-of-class). Concretely, we incorporate two in-class separation terms, defined as :

$$\ell_{\text{in}}^{(\mathcal{K})}(z) = \ln\big[\mathcal{K}(d_{\text{in}})\big] \quad \text{and} \quad \ell_{\text{out}}^{(\mathcal{K})}(z) = \ln\big[1 - \mathcal{K}(d_{\text{out}})\big].$$

The **In-class term** $\ell_{\text{in}}^{(\mathcal{K})}(z)$ rewards data points for remaining near their *correct* center, ensuring that normal or anomalous examples stay close to their respective centers. Conversely, the **out-of-class term** $\ell_{\text{out}}^{(\mathcal{K})}(z)$ penalizes points that approach the *wrong* center, driving them away from the wrong center.

**Lemma 6.1** (Light-Tailed Kernels for In-Class Compactness). *Suppose $\mathcal{K}_{\text{light}}$ decays very quickly at moderate distances, whereas $\mathcal{K}_{\text{heavy}}$ decays more slowly. Then for a point $z$ near or moderately far from its own class center $c_0$,*

$$\big\|\nabla_z \ell_{\text{in}}^{(\mathcal{K}_{\text{light}})}(z)\big\| > \big\|\nabla_z \ell_{\text{in}}^{(\mathcal{K}_{\text{heavy}})}(z)\big\|.$$

Consequently, $\mathcal{K}_{\text{light}}$ exerts a stronger "pull" toward $c_0$, resulting in a strictly tighter in-class cluster than $\mathcal{K}_{\text{heavy}}$.

In contrast, a *heavy-tailed* kernel better preserves a non-zero similarity at moderate distances, thereby maintaining an outward "push" that increases separation for out-of-class points.

**Lemma 6.2** (Heavy-Tailed Kernels for Out-of-Class Separation). *Assume $\mathcal{K}_{\text{light}}$ decays to nearly zero at moderate distances, whereas $\mathcal{K}_{\text{heavy}}$ remains non-negligible. Then for an out-of-class point $z$ with moderate $\|z - c_1\|$,*

$$\big\|\nabla_z \ell_{\text{out}}^{(\mathcal{K}_{\text{heavy}})}(z)\big\| > \big\|\nabla_z \ell_{\text{out}}^{(\mathcal{K}_{\text{light}})}(z)\big\|. \quad (7)$$

*Hence, $\mathcal{K}_{\text{heavy}}$ continues pushing out-of-class samples farther than $\mathcal{K}_{\text{light}}$, achieving a strictly larger separation margin.*

### 6.2. No Single Kernel Can Do Both

As established in Lemmas 6.1 and 6.2, a light-tailed kernel enforces tight in-class clustering but fails to maintain a wide out-of-class margin, whereas a heavy-tailed kernel preserves a broad margin but cannot strongly pull in-class points together. Consequently, no single kernel can satisfy both goals at once, motivating a dual-tailed-kernel approach.

**Theorem 6.3** (Dual-tailed Kernel Outperforms Single Kernel). *Let a single-kernel approach use the same function $\mathcal{K}_S$ for both $\ln[\mathcal{K}_S(d_{\text{in}})]$ (in-class term) and $\ln[1 - \mathcal{K}_S(d_{\text{out}})]$ (out-of-class term). Let a dual-tailed kernel approach instead use $\mathcal{K}_{\text{light}}$ for in-class and $\mathcal{K}_{\text{heavy}}$ for out-of-class. Under mild assumptions (e.g., sufficient model capacity or well-separated data), any single-kernel method must compromise either on compactness for in-class points or on maintaining a wide out-of-class margin. By contrast, the dual-tailed kernel design achieves both objectives simultaneously.*

The proof follows directly by Lemmas 6.1 and 6.2.

Based on these observations we define the dual-tailed kernel separation loss as:

**Dual-tailed kernel** ($\mathcal{K}_{\text{light}}, \mathcal{K}_{\text{heavy}}$) **Separation Loss.** Building on the insight of Theorem 6.3, we unify these findings into a ratio-based *logistic* form. Rather than summing a separate in-class reward $(\ln[\mathcal{K}_{\text{light}}(\cdot)])$ and out-of-class penalty $(\ln[1 - \mathcal{K}_{\text{heavy}}(\cdot)])$, we define the sample wise dual-tailed kernel loss as:

$$\ell_{\text{separation}}(\theta; x, y) = -\ln\left[\frac{\mathcal{K}_{\text{light}}(d_{\text{in}})}{\mathcal{K}_{\text{light}}(d_{\text{in}}) + \mathcal{K}_{\text{heavy}}(d_{\text{out}})}\right]. \quad (8)$$

and based on the wise loss, we define the total separation loss as:

$$\mathcal{L}_{\text{separation}}(\theta) = \underbrace{\sum_{(x,y)\in\mathcal{D}} \ell_{\text{separation}}(\theta; x, y)}_{\text{sum of single-sample losses}} \quad (9)$$

This encourages each sample to have higher similarity to its correct center (via the light-tailed kernel) than to the opposite center (heavy-tailed kernel). The ratio inside the logarithm naturally captures both *closeness to the correct center* (via the light-tailed term in the numerator) and *distance from the opposing center* (via the heavy-tailed term in the denominator). Numerically, this ratio-based design is stable, and conceptually, it aligns with a softmax-style interpretation, ensuring each sample is more "similar" to its own center than the other while leveraging the complementary strengths of light and heavy tails.

To quantify the benefit of using separate light-tailed and heavy-tailed kernels, we perform an ablation in Appendix K where we replace our dual-tailed design with a single kernel for both in-class and out-of-class distances. Table 8 below summarizes the main findings: We observe that the dual-tailed approach outperforms the single-kernel baseline. In particular, the heavy-tailed component preserves a more effective "push" for out-of-class samples, while the light-tailed component enforces tighter in-class clustering.

### 6.3. Necessity of Diversity Loss

Relying solely on the separation loss can yield degenerate solutions if the model is sufficiently expressive (e.g., large enough parameters). In such cases, the model can drive its training error near zero by mapping every instance of a class onto a single point in latent space, leading to poor generalization (Goyal et al., 2020).

**Lemma 6.4** (Degenerate Solutions with Fixed Centers). *If the network $f_\theta \colon \mathbb{R}^D \to \mathbb{R}^d$ is sufficiently expressive, then there exists a parameter set $\theta^*$ such that*

$$f_{\theta^*}(x) = \begin{cases} c_0, & \text{if } x \in \mathcal{U}, \\ c_1, & \text{if } x \in \mathcal{A}, \end{cases} \quad (10)$$

*and this degenerate mapping drives the training separation loss $\mathcal{L}_{\text{separation}}(\theta^*)$ arbitrarily close to 0.*

This result motivates the introduction of the **diversity term** (Eq. 11), which counteracts the tendency to collapse by fostering variability within each class. By penalizing over-concentration, the diversity term ensures that normal data retains subtle variations and that anomalies preserve their inherent heterogeneity, enhancing the model's robustness and generalization capabilities (Goyal et al., 2020). Specifically, for class $C \in \{\mathcal{A}, \mathcal{U}\}$, we define the average pairwise similarity:

$$k(C; \theta) = \frac{1}{|C|^2} \sum_{i,j\in C} \exp\left(-\frac{\|f_\theta(x_i)-f_\theta(x_j)\|}{\sigma_C^2}\right), \quad (11)$$

where $\sigma > 0$ is a fixed scale. The *diversity loss* is

$$\mathcal{L}_{\text{diversity}}(\theta) = k(\mathcal{A}; \theta) + k(\mathcal{U}; \theta). \quad (12)$$

A high value of $k(C; \theta)$ indicates that points in $C$ are mapped very close together in latent space (since the exponential term $\exp(-\|\cdot\|/\sigma^2)$ is close to 1). We *minimize* $\mathcal{L}_{\text{diversity}}$ to disfavor such overly tight clustering. A key advantage of this exponential form is its smoothly decaying gradient, which provides better control over moderate distances than linear or threshold-based metrics. Moreover, allowing separate $\sigma_C$ values for normal and anomalous classes captures their differing spread without adding significant complexity.

**Lemma 6.5** (Diversity Lower Bound for Collapsed Mapping). *Let $\mathcal{U}$ and $\mathcal{A}$ each contain at least two samples. Suppose $f_{\theta^*}$ is the fully collapsed mapping:*

$$f_{\theta^*}(x) = \begin{cases} \tilde{c}_0, & x \in \mathcal{U}, \\ \tilde{c}_1, & x \in \mathcal{A}, \end{cases} \quad \text{with } \tilde{c}_0 \neq \tilde{c}_1.$$

*Then*

$$k(\mathcal{U}; \theta^*) = 1, \quad k(\mathcal{A}; \theta^*) = 1, \quad \text{so} \quad \mathcal{L}_{\text{diversity}}(\theta^*) = 2.$$

Based on these insights we define the overall loss ass:

$$\mathcal{L}_{\text{total}}(\theta) = \mathcal{L}_{\text{separation}}(\theta) + \mathcal{L}_{\text{diversity}}(\theta), \quad (13)$$

**Corollary 6.6.** *If the total loss is*

$$\mathcal{L}_{\text{total}}(\theta) = \mathcal{L}_{\text{separation}}(\theta) + \mathcal{L}_{\text{diversity}}(\theta) \quad \text{with },$$

*then at the collapsed solution $\theta^*$ we have*

$$\mathcal{L}_{\text{total}}(\theta^*) = \mathcal{L}_{\text{separation}}(\theta^*) + 2 \geq 2.$$

*Thus no collapsed solution can have zero total loss. Hence, diversity prevents degeneracy.*

**Discussion** We want to point out that the degenerate arrangement does not globally minimize the total loss. Consider a non-degenerate mapping in which points lie close to their respective centers rather than coinciding exactly. By choosing this partial spread, we keep the separation loss $L_{\text{separation}}(\theta)$ arbitrarily small—say $\varepsilon$—while lowering the diversity loss $L_{\text{diversity}}(\theta)$ below 2 by some margin $\delta > 0$. In total we have:

$$\mathcal{L}_{\text{total}}(\theta) = \mathcal{L}_{\text{separation}}(\theta) + \mathcal{L}_{\text{diversity}}(\theta) \leq \varepsilon + (2-\delta) < 2.$$

Because the degenerate arrangement fixes the total loss at 2, any such non-degenerate solution with $\varepsilon < \delta$ strictly improves upon it. Consequently, the collapsed mapping is not optimal: **degeneracy cannot minimize $\mathcal{L}_{\text{total}}$.**

The separation loss pushes each sample toward its own center and away from the opposite center, while the diversity loss preserves intra-class variability to prevent collapse.

### 6.4. Computational Complexity of the Diversity Term.

A naive implementation of the diversity term incurs $\mathcal{O}(N^2)$ cost for a dataset of size $N$. We instead compute it *within* each mini-batch of size $b$, reducing the cost to $\mathcal{O}(b^2)$ per iteration. To go further, we *uniformly subsample* $b_s = \sqrt{b}$ points from each class within the batch, yielding a $\mathcal{O}(b_s^2) = \mathcal{O}(b)$ complexity per iteration. This strategy preserves the diversity penalty's effectiveness while keeping its overhead linear in the batch size.

## 7. Ensemble-Based Subset Splitting.

Let $\mathcal{D}_U$ be the unlabeled dataset and $\mathcal{D}_L$ the set of labeled anomalies. We partition $\mathcal{D}_U$ into $M$ disjoint subsets, $\{\mathcal{D}_U^{(m)}\}_{m=1}^M$, by splitting the data indices into $M$ consecutive blocks. As a result,

$$\bigcup_{m=1}^M \mathcal{D}_U^{(m)} = \mathcal{D}_U, \quad \mathcal{D}_U^{(i)} \cap \mathcal{D}_U^{(j)} = \varnothing \quad \text{for } i \neq j. \quad (14)$$

Note that if $|\mathcal{D}_U|$ is not exactly divisible by $M$, the last subset may contain fewer samples, but altogether they still cover $\mathcal{D}_U$ fully without overlap. Each subset $\mathcal{D}_U^{(m)}$ is then *combined* with the same set of labeled anomalies $\mathcal{D}_L$ to form a training set for the $m$-th model:

$$\mathcal{D}^{(m)} = \mathcal{D}_U^{(m)} \cup \mathcal{D}_L.$$

Hence, every model in the ensemble sees a *unique* slice of unlabeled data while sharing the same anomalies.

**Training Ensemble Components.** Denote by $f_m(\cdot)$ the feature mapping or anomaly score network trained on the set $\mathcal{D}^{(m)}$. Let $\Theta_m$ be its parameters, learned by minimizing a suitable objective (Eq. 13):

$$\Theta_m^* = \arg\min_{\Theta_m} \mathcal{L}\big(\Theta_m; \mathcal{D}_U^{(m)}, \mathcal{D}_L\big), \quad m = 1, 2, \ldots, M.$$

Since all $\mathcal{D}^{(m)}$ share $\mathcal{D}_L$, each model is exposed to the same limited but crucial anomaly examples, preventing any single model from entirely ignoring the labeled anomaly information.

### 7.1. Aggregating Ensemble Outputs.

During inference, a test point $x$ is passed through each trained model, and the anomaly score $\phi_m(x)$ for the $m$-th

model is computed as:

$$\phi_m(x) = 1 - \frac{\mathcal{K}_{\text{heavy}}\big(\|r_{m,0}(x)\|\big)}{\mathcal{K}_{\text{heavy}(x)}\big(\|r_{m,0}(x)\|\big) + \mathcal{K}_{\text{heavy}}\big(\|r_{m,1}\|\big)}. \quad (15)$$

where we define

$$r_{m,0}(x) = f_{\Theta_m}(x) - c_0 \quad \text{and} \quad r_{m,1}(x) = f_{\Theta_m}(x) - c_1.$$

$\phi_m(x)$ is bounded between [0,1]. For $\phi_m$ we utilize the heavy-tailed kernel, due to the diversity term, normal points naturally spread around $c_0$, but still remain closer than anomalies. A light-tailed kernel would penalize even moderate distances and risk mislabeling these normal points. Instead, the heavy-tailed kernel in Eq. 15 retains sufficient similarity at moderate distances, ensuring normal points consistently receive lower anomaly scores, while anomalies lying farther out are assigned higher scores. We then aggregate the $M$ scores into a single final score by averaging:

$$\phi(x) = \frac{1}{M} \sum_{m=1}^M \phi_m(x). \quad (16)$$

The anomaly score $\phi(x)$ is between [0,1]. Anomalies have an anomaly score of $\approx 1$, while normal samples have a significantly lower anomaly score of $\approx 0$.

**Discussion.** Splitting unlabeled data gives each ensemble model a distinct view of normality, while sharing the same anomaly labels ensures consistent guidance. Aggregating these diverse detectors improves robustness and generalization under limited anomaly labels. Appendix L provides an ablation study on ensemble size. Our empirical sweep (Appendix L) shows that $M = 5$ strikes a good balance—achieving near-optimal performance gains over smaller ensembles (like $M = 1, 3$) without incurring the heavy computational cost of even larger ensembles. Hence, we fix $M = 5$ as our default throughout the experiments.

## 8. Experiments

### 8.1. Experimental Setup

We compare WSAD-DT with state-of-the-art deep anomaly detection methods on over 20 real-world datasets from the AdBenchmark repository (Han et al., 2022). Each dataset is split into 70% training and 30% testing, preserving the anomaly ratio via stratified sampling. In this experiment, we label only 5% of anomalies (or 5 anomalies, whichever is larger), chosen uniformly at random from the training set, ensuring minimal yet consistent supervision across all runs. Note that in these main experiments, we *do not* add contamination in the training data; the limited anomaly labels ($\leq 5\%$) reflect our focus on weaker supervision (in

*Table 1.* This table presents the results of all algorithms using the default parameters outlined in the original paper. Hereby, the best values are shown in bold, and the runner-up is underlined. The 'AVG Rank' row of the table lists the average rank achieved by all algorithms in the metric AUC-ROC. The lower the rank, the better the result. The last row of the table contains the adjusted p-value of the Wilcoxon signed-rank test at an alpha = 0.05 for comparison between the WSAD-DT and reference methods. The symbol '+' represents situations where the WSAD-DT is statistically superior to the comparing method.

| Dataset | WSAD-DT | DeepSAD | DevNet | FeaWAD | GANAnomaly | PreNet | ROSAS | XGBOD |
|---|---|---|---|---|---|---|---|---|
| Optdigits | 0.9996 (3) | 0.9773 (7) | 0.9959 (4) | 0.9839 (5) | 0.6041 (8) | **1.0000 (1)** | **1.0000 (1)** | 0.9817 (6) |
| Lymphography | **1.0000 (1)** | **1.0000 (1)** | 0.9961 (4) | 0.9787 (6) | 0.8953 (7) | **1.0000 (1)** | ∗ | 0.9903 (5) |
| Pendigits | **0.9998 (1)** | 0.9768 (5) | 0.9682 (6) | 0.7798 (8) | 0.8361 (7) | 0.9832 (4) | 0.9997 (2) | 0.9981 (3) |
| Vertebral | **0.9051 (1)** | 0.7810 (5) | 0.4974 (7) | 0.6399 (6) | 0.3618 (8) | 0.9036 (2) | 0.8377 (3) | 0.8125 (4) |
| Wdbc | **1.0000 (1)** | 0.9990 (3) | **1.0000 (1)** | 0.9979 (4) | 0.9830 (8) | 0.9969 (6) | 0.9933 (7) | 0.9979 (4) |
| Wpbc | **0.6915 (1)** | 0.6144 (4) | 0.6403 (3) | 0.5619 (6) | 0.4635 (7) | 0.6649 (2) | ∗ | 0.6135 (5) |
| Stamps | **0.9902 (1)** | 0.9492 (4) | 0.8909 (6) | 0.8664 (7) | 0.7529 (8) | 0.9697 (2) | 0.9544 (3) | 0.9011 (5) |
| Satimage-2 | **0.9973 (1)** | 0.9715 (5) | 0.9562 (6) | 0.9753 (4) | 0.9783 (3) | 0.9243 (8) | 0.9526 (7) | 0.9866 (2) |
| Spambase | 0.9499 (2) | 0.8669 (5) | 0.9389 (3) | ∗ | 0.6108 (7) | 0.8453 (6) | 0.9128 (4) | **0.9610 (1)** |
| Thyroid | 0.9960 (2) | 0.9782 (4) | 0.9667 (5) | 0.8525 (7) | 0.7795 (8) | 0.9420 (6) | **0.9961 (1)** | 0.9795 (3) |
| Mnist | **0.9883 (1)** | 0.9116 (4) | 0.9037 (6) | 0.8100 (8) | 0.8127 (7) | 0.9093 (5) | 0.9609 (3) | 0.9870 (2) |
| Yeast | **0.6545 (1)** | 0.6336 (3) | 0.5942 (5) | 0.5360 (7) | 0.4867 (8) | 0.6309 (4) | 0.6521 (2) | 0.5922 (6) |
| Cardio | **0.9908 (1)** | 0.9842 (3) | 0.9778 (4) | 0.8067 (8) | 0.9084 (7) | 0.9422 (6) | 0.9567 (5) | 0.9904 (2) |
| Vowels | **0.9736 (1)** | 0.9712 (2) | 0.8839 (6) | 0.7696 (8) | 0.8231 (7) | 0.9300 (5) | 0.9653 (3) | 0.9516 (4) |
| Wine | **1.0000 (1)** | **1.0000 (1)** | 0.9954 (5) | 0.9491 (6) | 0.6836 (7) | **1.0000 (1)** | ∗ | **1.0000 (1)** |
| Magic.gamma | **0.9166 (1)** | 0.8719 (4) | 0.8282 (5) | 0.7228 (7) | 0.6817 (8) | 0.8260 (6) | 0.9034 (3) | 0.9159 (2) |
| Ionosphere | **0.9779 (1)** | 0.9704 (3) | 0.5232 (7) | 0.4238 (8) | 0.6880 (6) | 0.7116 (5) | 0.7656 (4) | 0.9746 (2) |
| Glass | **0.9830 (1)** | 0.9355 (3) | 0.8199 (6) | 0.6891 (8) | 0.7106 (7) | 0.8934 (5) | 0.8952 (4) | 0.9606 (2) |
| Breastw | 0.9894 (2) | 0.9508 (5) | **0.9947 (1)** | ∗ | 0.9559 (4) | 0.9402 (6) | 0.7851 (7) | 0.9842 (3) |
| Yelp | **0.8311 (1)** | 0.7139 (4) | 0.6812 (5) | 0.5522 (8) | 0.6567 (7) | 0.6670 (6) | 0.7950 (2) | 0.7584 (3) |
| Imdb | **0.7623 (1)** | 0.6730 (3) | 0.6204 (5) | 0.5683 (6) | 0.5021 (8) | 0.5464 (7) | 0.6473 (4) | 0.6996 (2) |
| MNIST-C-Fog | **1.0000 (1)** | **1.0000 (1)** | **1.0000 (1)** | 0.9529 (7) | 0.8012 (8) | **1.0000 (1)** | **1.0000 (1)** | 0.9998 (6) |
| MNIST-C-canny-Edges | **0.9999 (1)** | 0.9859 (6) | 0.9927 (5) | 0.8818 (7) | 0.6937 (8) | 0.9932 (4) | 0.9996 (2) | 0.9982 (3) |
| MVTec-AD-Zipper | **0.9307 (1)** | 0.8626 (3) | 0.7728 (6) | 0.5132 (8) | 0.7402 (7) | 0.7815 (5) | 0.8993 (2) | 0.8455 (4) |
| MNIST-C-Stripe | **1.0000 (1)** | **1.0000 (1)** | **1.0000 (1)** | 0.9697 (8) | 0.9733 (7) | **1.0000 (1)** | **1.0000 (1)** | **1.0000 (1)** |
| Skin | **0.9998 (1)** | 0.9995 (2) | 0.9937 (5) | ∗ | 0.5202 (6) | † | 0.9992 (4) | 0.9995 (2) |
| Fraud | 0.9574 (2) | 0.9504 (4) | 0.9189 (6) | 0.7855 (7) | 0.9300 (5) | † | 0.9521 (3) | **0.9612 (1)** |
| Http | **1.0000 (1)** | **1.0000 (1)** | 0.9984 (4) | ∗ | 0.7011 (6) | † | 0.9979 (5) | 0.9997 (3) |
| Cover | 0.9996 (2) | 0.9976 (4) | 0.9992 (3) | 0.6304 (6) | 0.4702 (7) | † | **0.9997 (1)** | 0.9946 (5) |
| Shuttle | 0.9956 (2) | 0.9942 (3) | 0.9775 (6) | 0.9717 (7) | 0.9781 (5) | † | 0.9864 (4) | **0.9993 (1)** |
| AVGRank | 1.27 | 3.43 | 4.57 | 6.90 | 6.87 | 4.77 | 3.73 | 3.10 |
| p-value | N/A | 0.00022220 (+) | 0.00020919 (+) | 0.00000005 (+) | 0.00000005 (+) | 0.00019809 (+) | 0.00019916 (+) | 0.00344150 (+) |

Values marked with † indicate that no result was available within 12 hours.
Values marked with ∗ indicate that a runtime error occurred during execution.

line with (Han et al., 2022; Pang et al., 2023)). For an ablation study on contamination in the training data, see Appendix M. Each experiment is repeated on three splits and averaged, with performance measured by AUC-ROC and AUC-PR (Davis & Goadrich, 2006) and significance tested via paired Wilcoxon signed-rank (Holm-Bonferroni correction (McDonald, 2014)). All features are scaled to $[0, 1]$ (MinMaxScaler (Pedregosa et al., 2011)); five seeds ($\{0,1,2,100,1000\}$) are used for stochastic methods; and default hyperparameters are taken from the original references (Appendix I). We set the ensemble size to 5 (Ablation study for ensemble size see Appendix L), and provide further dataset statistics, kernel choices, parameter details, and experiments with varied labeled-anomaly proportions in Appendices H and N.

## 8.2. Real-world data

Table 1 and the Appendix F summarize the AUC-ROC and AUC-PR results across all competing methods. WSAD-DT ranks *first* on 23 datasets and *second* on 6 for AUC-ROC (Table 1), attaining an average rank of 1.27. By contrast, baselines such as DeepSAD and DevNet—relying on single centers or single-tail distributions—perform less effectively. WSAD-DT also excels in AUC-PR (Appendix F), ranking *first* on 15 datasets and *second* on 11, with an average rank of 1.70. Wilcoxon signed-rank tests confirm these improvements are statistically significant. The gains primarily arise from WSAD-DT's dual-tailed kernel, which integrates light- and heavy-tailed similarities for tighter in-class clustering and a broader inter-class margin, along with its diversity term that prevents trivial collapse. Moreover, WSAD-DT's ensemble—splitting unlabeled data but sharing few labeled anomalies—outperforms gradient-boosting ensembles (e.g., XGBOD), highlighting the benefits of combining scarce anomaly labels with diverse unlabeled partitions. Even under contamination in the unlabeled dataset, WSAD-DT maintains strong performance across diverse benchmarks. To assess the robustness of WSAD-DT, Appendix M presents additional experiments simulating varying contamination rates. Our results show that WSAD-DT maintains strong performance even as unlabeled data becomes increasingly corrupted. In particular, it degrades gracefully compared to other methods, underscoring the resilience of the dual-tailed kernel design and ensemble approach under noisy supervision. Moreover, it shows resilience to hyperparameter choices, reinforcing its reliability in real-world scenarios with varying data conditions.

## 9. Conclusion

We introduced WSAD-DT, a dual-tailed kernel framework for weakly supervised anomaly detection that employs a light-tailed kernel for in-class compactness and a heavy-

tailed kernel for robust out-of-class separation. This design allows WSAD-DT to learn a latent space where normal and anomalous samples are effectively distinguished, aided by a diversity term that prevents degenerate all-points-collapse mappings. In addition, we split the unlabeled dataset into multiple partitions while sharing a small set of labeled anomalies across them; the resulting ensemble achieves state-of-the-art results under limited supervision. Our dual-kernel similarity approach resolves the tension between tight clustering and wide margins, outperforming single-kernel methods in both in-class pull and out-of-class push. Splitting unlabeled data into multiple partitions enables each model to develop a different view of normality, while shared anomaly labels enforce a consistent notion of anomaly. An ensemble size of $M = 5$ was found to offer a practical trade-off between accuracy and computational cost. Moreover, the method demonstrates strong robustness to weak labels, effectively identifying anomalies using as few as five labeled anomalies across various datasets.

## 10. Future Directions

Although WSAD-DT excels on static tabular data, it does not explicitly handle temporal or relational structures. Real-world tasks (e.g., fraud or sensor monitoring) often involve time-series or graph data, where anomalies manifest through evolving patterns or network connections. Adapting WSAD-DT to recurrent or transformer-based models (for sequences) or graph neural networks (for relational data) could significantly broaden its applicability.

## Impact Statement

This paper presents work whose goal is to advance the field of Machine Learning. There are many potential societal consequences of our work, none of which we feel must be specifically highlighted here.

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

| Appendix Section | Content |
|---|---|
| Appendix A | Notation summary |
| Appendix B | Proof of Lemma 6.1 |
| Appendix C | Proof of Lemma 6.2 |
| Appendix D | Proof of Lemma 6.4 |
| Appendix E | Proof of Lemma 6.5 |
| Appendix F | AUC-PR results |
| Appendix G | Algorithms Details |
| Appendix H | Experiment details |
| Appendix I | Implementation Details |
| Appendix J | Scalability Test |
| Appendix K | Ablation study light-tailed, heavy-tailed, dual-tailed kernel |
| Appendix M | Analyzing the performance under contamination in the training data |
| Appendix L | Ablation study on the effect of different numbers of ensembles |
| Appendix N | Ablation study for different numbers of labeled anomalies |
| Appendix O | Parameter Sensitivity |
| Appendix P | Ablation study kernel regularization |
| Appendix Q | Limitation |

*Table 2.* Structure of our appendix.

## A. Notation summary

Table 3 provides a summary of the notation used throughout this paper.

*Table 3.* Notation Summary

| Symbol | Description |
|---|---|
| $\mathcal{D}_U$ | Unlabeled dataset (primarily normal but may be contaminated) |
| $\mathcal{D}_L$ | Labeled set of anomalies |
| $N, K$ | Number of unlabeled points ($N$), number of labeled anomalies ($K$) |
| $x_i \in \mathbb{R}^D$ | $i$-th input data point (feature vector of dimension $D$) |
| $y_i \in \{0, 1\}$ | Binary label indicating normal ($y = 0$) or anomaly ($y = 1$) |
| $f_\theta(\cdot)$ | Neural network mapping from input space to latent space |
| $z_i = f_\theta(x_i)$ | Latent representation (embedding) of point $x_i$ |
| $c_0, c_1 \in \mathbb{R}^d$ | Centers in latent space for normal ($c_0$) and anomalous ($c_1$) classes |
| $d_{i,k}$ | Distance between $z_i$ and center $c_k$, e.g. $\|z_i - c_k\|^2$ |
| $d_{\text{in}}, d_{\text{out}}$ | In-class vs. out-of-class distance for a sample, depending on $y_i$ |
| $\mathcal{K}_{\text{light}}(d)$ | Light-tailed kernel (e.g. Gaussian), for in-class distances |
| $\mathcal{K}_{\text{heavy}}(d)$ | Heavy-tailed kernel (e.g. Student-$t$), for out-of-class distances |
| $\sigma_{\mathcal{U}}, \sigma_{\mathcal{A}}$ | Bandwidth parameters for the light-tailed kernels (normal/anomaly) |
| $\nu$ | Degrees-of-freedom parameter for the heavy-tailed kernel |
| $\mathcal{L}_{\text{separation}}(\theta)$ | Separation loss term |
| $\mathcal{L}_{\text{diversity}}(\theta)$ | Diversity (regularization) loss term |
| $\mathcal{L}_{\text{total}}(\theta)$ | Overall training objective, sum of separation + diversity |
| $\phi_m(x)$ | Anomaly score output by the $m$-th model in the ensemble |
| $\phi(x)$ | Final ensemble anomaly score (aggregation of $\phi_m$) |

# B. Proof Lemma 6.1

*Proof.*     1. **General Gradient Form.** For $\ell_{\text{in}}^{(\mathcal{K})}(z) = \ln[\mathcal{K}(d)]$ with $d = \|z - c_0\|$, we compute:

$$\nabla_z \ell_{\text{in}}^{(\mathcal{K})}(z) = \frac{d\,\mathcal{K}(d)}{d\,(d)} \cdot \frac{1}{\mathcal{K}(d)} \cdot \nabla_z\big(\|z - c_0\|\big).$$

Since

$$\nabla_z \|z - c_0\| = \frac{z - c_0}{\|z - c_0\|}.$$

, we have

$$\nabla_z \ell_{\text{in}}^{(\mathcal{K})}(z) = \frac{\mathcal{K}'(d)}{\mathcal{K}(d)} \cdot \frac{z - c_0}{\|z - c_0\|}.$$

Its magnitude is

$$\|\nabla_z \ell_{\text{in}}^{(\mathcal{K})}(z)\| = \big|\tfrac{\mathcal{K}'(d)}{\mathcal{K}(d)}\big|.$$

2. **Moderate to large $\|z - c_0\|$: $\mathcal{K}_{\text{light}}$ is smaller $\Rightarrow$ bigger reciprocal.**

$$\left\|\nabla_z \ell_{\text{in}}^{(\mathcal{K}_{\text{light}})}(z)\right\| \gg \left\|\nabla_z \ell_{\text{in}}^{(\mathcal{K}_{\text{heavy}}^{(\text{only})})}(z)\right\| \quad \text{for large } d.$$

Because $\mathcal{K}_{\text{light}}(d) \sim e^{-\alpha d}$, we get

$$\mathcal{K}'_{\text{light}}(d) \approx -\alpha\, e^{-\alpha d}.$$

Thus

$$\frac{\mathcal{K}'_{\text{light}}(d)}{\mathcal{K}_{\text{light}}(d)} = \frac{-\alpha\, e^{-\alpha d}}{e^{-\alpha d}} = -\alpha,$$

a *constant in magnitude*. Meanwhile, $\mathcal{K}_{\text{light}}(d)$ itself is extremely small for large $d$, making $\ln[\mathcal{K}_{\text{light}}(d)]$ strongly negative and its gradient *large in magnitude*. This yields a **strong inward pull** for any in-class point at moderate/large distance

If we replaced $\mathcal{K}_{\text{light}}$ by a heavier kernel $\mathcal{K}_{\text{heavy}} \sim d^{-\beta}$, then at moderate $d$ we have

$$\mathcal{K}'_{\text{heavy}}(d) \sim (-\beta d^{-\beta-1}), \quad \frac{\mathcal{K}'_{\text{heavy}}(d)}{\mathcal{K}_{\text{heavy}}(d)} \sim -\frac{\beta}{d} \to 0.$$

This smaller ratio translates to a weaker inward gradient. Therefore, $\mathcal{K}_{\text{light}}$ is strictly more "aggressive" about pulling in-class points near $c_0$.

Thus, the light-tailed kernel exerts a stronger inward pull, preventing normal points from staying at large distances from the true center.

Because the $\mathcal{K}_{\text{light}}$ gradient strictly exceeds that of $\mathcal{K}_{\text{heavy}}^{(\text{only})}$ for moderate to large distances, it *never* tolerates a normal point lingering far from $c_0$. Consequently, the final in-class cluster is strictly tighter under $\mathcal{K}_{\text{light}}$.

$\square$

**Remark 1.** In a deep model, $z = f_\theta(x)$ depends on parameters $\theta$. Although the proof here focuses on $\nabla_z \ell$, the actual training updates weights via chain rule:

$$\nabla_\theta\, \ell_{\text{in}}(\theta) = \nabla_z \ell_{\text{in}} \cdot \nabla_\theta z = \big(\nabla_z \ell_{\text{in}}\big) \times \big(\nabla_\theta f_\theta(x)\big).$$

Since a larger $\|\nabla_z \ell\|$ directly contributes to a larger $\|\nabla_\theta \ell\|$, the strong "pull" evident in the latent space indeed translates into stronger parameter updates in backpropagation. Therefore, if $\mathcal{K}_{\text{light}}$ induces a bigger inward gradient on $z$, it also induces larger weight updates, more tightly clustering in-class points around $c_0$.

**Remark 2.** An identical argument applies if we replace $c_0$ by another class center $c_1$. The same exponential vs. polynomial decay comparison shows that a light-tailed kernel $\mathcal{K}_{\text{light}}$ always exerts a stronger inward pull, regardless of which center is in question.

# C. Proof Lemma 6.2

*Proof.* Let $d = \|z - c_1\|$. Then:

$$\ell_{\text{out}}^{(\mathcal{K})}(z) = \ln\big[\, 1 - \mathcal{K}(d)\,\big], \quad \nabla_z \ell_{\text{out}}^{(\mathcal{K})}(z) = \frac{1}{1 - \mathcal{K}(d)}\,\big(-\mathcal{K}'(d)\big)\,\cdot\,\frac{z - c_1}{\|z - c_1\|}.$$

Focus on moderate $d$: We compute the gradient of $\ell_{\text{out}}^{(\mathcal{K})}(z)$ with respect to $z$. Using the chain rule:

$$\nabla_z \ell_{\text{out}}^{(\mathcal{K})}(z) = \frac{\partial}{\partial z} \ln\big[\, 1 - \mathcal{K}(d)\,\big].$$

Let $d = \|z - c_1\|^2$. Then:

$$\frac{\partial}{\partial d} \ln\big[\, 1 - \mathcal{K}(d)\,\big] = \frac{1}{1 - \mathcal{K}(d)}\big[-\mathcal{K}'(d)\big],$$

and

$$\frac{\partial(d)}{\partial z} = \frac{\partial}{\partial z}\|z - c_1\| = \frac{z - c_0}{\|z - c_0\|}.$$

Thus:

$$\nabla_z \ell_{\text{out}}^{(\mathcal{K})}(z) = \frac{-\mathcal{K}'(d)}{1 - \mathcal{K}(d)}\,\cdot\,\frac{z - c_1}{\|z - c_1\|}.$$

Note that $\mathcal{K}'(d)$ is the derivative of $\mathcal{K}$ with respect to $\|z - c_1\|$.

**3. The behavior of Light-Tailed vs. Heavy-Tailed Kernels at Moderate Distances**    (a) **Light-Tailed Kernel** $\mathcal{K}_{\text{light}}$: By definition, a light-tailed kernel rapidly decays toward (or extremely close to) zero when $d$ is only moderately large. Hence, for moderate $d$, we typically have:

$$\mathcal{K}_{\text{light}}(d) \approx 0, \quad \mathcal{K}'_{\text{light}}(d) \approx 0.$$

Substituting into the gradient formula:

$$\nabla_z \ell_{\text{out}}^{(\mathcal{K}_{\text{light}})}(z) \approx \frac{-0}{1 - 0}\,\cdot\,\frac{z - c_1}{\|z - c_1\|} = 0.$$

Consequently, the out-of-class log-loss $\ln[1 - \mathcal{K}_{\text{light}}(d)]$ saturates at $\ln(1) = 0$, and the gradient becomes nearly zero. This implies there is almost *no incentive* to push $z$ farther away from $c_1$.

(b) **Heavy-Tailed Kernel** $\mathcal{K}_{\text{heavy}}$: A heavy-tailed kernel retains moderate positive values even at moderate distances. For such $d$, we might have:

$$\mathcal{K}_{\text{heavy}}(d) \neq 0, \quad \mathcal{K}'_{\text{heavy}}(d) \neq 0.$$

Then:

$$\nabla_z \ell_{\text{out}}^{(\mathcal{K}_{\text{heavy}})}(z) \approx \frac{-\mathcal{K}'_{\text{heavy}}(d)}{\mathcal{K}_{\text{heavy}}(d)}\,\cdot\,\frac{z - c_1}{\|z - c_1\|} \neq 0.$$

This gradient remains non-trivial, meaning the loss is *still increasing* as $z$ moves outward. The model thus continues to "push" $z$ away from $c_1$, promoting a wider separation boundary. $\qquad\square$

**Conclusion**    Because $\mathcal{K}_{\text{light}}$ effectively becomes negligible at moderate distances, $\ell_{\text{out}}^{(\mathcal{K}_{\text{light}})}$ saturates and yields no gradient for pushing out-of-class points farther. In contrast, $\mathcal{K}_{\text{heavy}}$ preserves a moderate value at similar distances, ensuring $\ell_{\text{out}}^{(\mathcal{K}_{\text{heavy}})}$ continues to provide a meaningful gradient that drives $z$ away from $c_1$.

**Remark 1**. Again, if $z = f_\theta(x)$ is computed by a neural net with parameters $\theta$, then standard backpropagation yields:

$$\nabla_\theta \ell_{\text{out}}(\theta) \;=\; \nabla_z \ell_{\text{out}} \;\times\; \nabla_\theta z.$$

Hence a sustained non-zero gradient in $\nabla_z \ell_{\text{out}}$ (under heavy tails) also implies a sustained non-zero gradient in $\nabla_\theta \ell_{\text{out}}$, causing the network weights to keep shifting $z$ away from $c_1$. By contrast, a light-tailed kernel's near-zero $\nabla_z$ at moderate $d$ translates into minimal weight updates once $z$ is already somewhat far from $c_1$.

**Remark 2.**The same argument applies if we swap centers. We specifically showed how a light-tailed kernel saturates when pushing $z$ away from $c_1$; however, one could equally consider pushing a point away from $c_0$ if it is out-of-class for that center. The key principle remains: a heavy-tailed kernel continues to produce a nonzero gradient at larger distances, while a light-tailed kernel saturates and stops pushing.

**Remark (Moderate Distances).**    Throughout Lemmas 5.1 and 5.2, we say a sample $z$ is at a *moderate distance* from a center $c_k$ if $\|z - c_k\|$ is neither so small as to be effectively zero, nor so large that the kernel has saturated at (or near) zero. Concretely, let $\delta_{\min}, \delta_{\max} > 0$ be thresholds such that

$$\delta_{\min} \; < \; \|z - c_k\| \; < \; \delta_{\max}.$$

- **Near Distance** ($\|z - c_k\| \approx 0$) leads to a similarity $\mathcal{K}(\|z - c_k\|) \approx 1$.

- **Large Distance** ($\|z - c_k\| \gg 1$) often results in $\mathcal{K}(\|z - c_k\|) \approx 0$ for a light-tailed kernel (or near saturation for a heavy-tailed kernel).

- **Moderate Distance** is precisely the range where the kernel's decay profile (exponential vs. polynomial) meaningfully differs and produces nontrivial gradient behavior.

In practice, "moderate distance" captures the radius at which a light-tailed kernel begins to drop off sharply, or a heavy-tailed kernel remains significantly above zero. It is in this regime that Lemmas 5.1 and 5.2 highlight how light-tailed vs. heavy-tailed kernels yield distinct advantages for in-class compactness and out-of-class separation, respectively.

**Experiment validation**    To further validate these observations, we conducted experiments on some real-world datasets, training each model for 100 epochs and reporting the results in Table 4. Our analysis focused on the average distances of anomalous and normal points to both the anomaly center ($c_1$) and the normal center ($c_0$), as detailed below:

$$i.a. \; := \; \frac{1}{|\mathcal{A}_{\text{in}}|} \sum_{x \in \mathcal{A}_{\text{in}}} d(x, c_1), \quad o.a. \; := \; \frac{1}{|\mathcal{A}_{\text{out}}|} \sum_{x \in \mathcal{A}_{\text{out}}} d(x, c_0),$$

$$i.n. \; := \; \frac{1}{|\mathcal{U}_{\text{in}}|} \sum_{x \in \mathcal{U}_{\text{in}}} d(x, c_0), \quad o.n. \; := \; \frac{1}{|\mathcal{U}_{\text{out}}|} \sum_{x \in \mathcal{U}_{\text{out}}} d(x, c_1).$$

Here:

- $\mathcal{A}_{\text{in}}$ and $\mathcal{U}_{\text{in}}$ denote points correctly assigned to their respective centers (anomalous to $c_1$, normal to $c_0$).

- $\mathcal{A}_{\text{out}}$ and $\mathcal{U}_{\text{out}}$ refer to points assessed in relation to the center of the opposite class.

**Key Findings**

**In-Class Distances (*i.a.* and *i.n.*:)**    The light-tailed kernel achieves significantly smaller in-class distances compared to the heavy-tailed kernel. This indicates that the light-tailed kernel promotes tighter clustering around the respective centers, confirming its ability to enforce compactness for in-class points.

**Out-of-Class Distances (*o.a.* and *o.n.*:)**    The heavy-tailed kernel results in larger out-of-class distances, effectively separating points from the center of the opposing class. This observation aligns with the heavy-tailed kernel's design, which preserves moderate similarity for points farther away, thereby maintaining a wider margin between classes.

**Implications**

The results validate the complementary roles of the light and heavy-tailed kernels as described in Lemmas 6.1 and 6.2. Specifically:

- The light-tailed kernel excels at compacting in-class points, ensuring that normal and anomalous points cluster tightly around their respective centers.

- The heavy-tailed kernel enforces broader margins, preventing out-of-class points from being falsely pulled toward the wrong center.

By leveraging both kernels, the dual-tailed kernel approach achieves robust in-class compactness and out-of-class separation, which are critical for distinguishing anomalies from normal points. This dual mechanism addresses the limitations of single-kernel methods and demonstrates superior adaptability across diverse datasets (see ablation study in Appendix K).

*Table 4.* Comparison of Average Distances Using Light-Tailed vs. Heavy-Tailed Kernels

| **Dataset** | Light-Tailed Kernel | | | | Heavy-Tailed Kernel | | | |
|---|---|---|---|---|---|---|---|---|
| | i.a. | o.a. | i.n. | o.n. | i.a. | o.a. | i.n. | o.n. |
| Vertebral | 0.3556 | 0.5176 | 0.4038 | 0.4495 | 0.8766 | 1.0620 | 0.9960 | 1.0299 |
| WDBC | 0.0568 | 1.1348 | 0.1310 | 1.2099 | 0.2737 | 1.4539 | 0.2862 | 1.4368 |
| WPBC | 0.1294 | 0.5579 | 0.3189 | 0.6069 | 0.5693 | 0.9703 | 0.9013 | 1.1783 |
| Stamps | 0.0588 | 0.8711 | 0.1115 | 0.8529 | 0.2683 | 1.0988 | 0.4663 | 1.2703 |
| Satimage-2 | 0.0274 | 0.9126 | 0.0581 | 0.9499 | 0.2904 | 1.2005 | 0.3640 | 1.2727 |
| Magic.gamma | 0.2421 | 0.5414 | 0.1981 | 0.6531 | 0.7599 | 1.1767 | 0.6396 | 1.1107 |
| Yeast | 0.2427 | 0.5069 | 0.3010 | 0.4875 | 0.7474 | 1.0097 | 0.8511 | 0.9507 |
| Vowels | 0.0363 | 0.6670 | 0.1455 | 0.7584 | 0.2462 | 0.8767 | 0.7248 | 1.3530 |
| Ionosphere | 0.0960 | 0.8152 | 0.1026 | 0.7668 | 0.3943 | 1.1142 | 0.5042 | 1.2127 |

# D. Proof of Lemma 6.4

*Proof.* **Zero Separation Loss.** The per-sample separation loss for a point $(x, y)$ takes the form

$$\ell_{\text{separation}}(\theta; x, y) = -\ln\left[\frac{\mathcal{K}_{\text{light}}\left(\|f_\theta(x) - c_{\text{in}}\|\right)}{\mathcal{K}_{\text{light}}\left(\|f_\theta(x) - c_{\text{in}}\|\right) + \mathcal{K}_{\text{heavy}}\left(\|f_\theta(x) - c_{\text{out}}\|\right)}\right],$$

where $c_{\text{in}}$ is the "correct" center for $x$ (i.e., $c_0$ if $y = 0$, or $c_1$ if $y = 1$), and $c_{\text{out}}$ is the opposite center.

Consider the mapping

$$f_{\theta^*}(x) = \begin{cases} c_0, & y = 0, \\ c_1, & y = 1. \end{cases}$$

Then $\|f_{\theta^*}(x) - c_{\text{in}}\| = 0$. Since $\mathcal{K}_{\text{light}}(0) \approx 1$ and $\|c_0 - c_1\| > 0$, the ratio inside the logarithm approaches 1, making $\ell_{\text{separation}}(\theta^*; x, y) \approx 0$ for each sample. Summing over the entire training set, $\mathcal{L}_{\text{separation}}(\theta^*) \approx 0$.

**Class Collapse and Consequences.** All normal points ($y = 0$) are mapped to $c_0$, and all anomalous points ($y = 1$) are mapped to $c_1$. Hence, each class is collapsed onto a single coordinate in the latent space, destroying any intra-class variability. While it yields "perfect" separation on the training set, this degenerate solution generalizes poorly (Goyal et al., 2020).

**Conclusion.** With fixed $c_0$ and $c_1$, a sufficiently flexible $f_\theta$ can achieve near-zero training separation loss by collapsing each class onto its respective center. This justifies the need for additional regularization (e.g., a diversity term) to preserve the intra-class structure and avoid trivial collapse. $\qquad\square$

# E. Proof of Lemma 6.5

*Proof.* Fix $\theta^*$ such that $f_{\theta^*}(x)$ is constant on $\mathcal{U}$. Pick any $x_i, x_j \in \mathcal{U}$. Then $\|f_{\theta^*}(x_i) - f_{\theta^*}(x_j)\| = \|\tilde{c}_0 - \tilde{c}_0\| = 0$, so the exponential similarity $\exp(-\|z_i - z_j\|/\sigma^2) = \exp(0) = 1$. Hence every pair $(i, j) \in \mathcal{U} \times \mathcal{U}$ contributes 1. Summing over $|\mathcal{U}|^2$ pairs and dividing by $|\mathcal{U}|^2$ yields $k(\mathcal{U}; \theta^*) = 1$. Identical reasoning applies to $\mathcal{A}$. A summation of the two completes the proof. $\qquad\square$

**Corollary:** If the total loss is $\mathcal{L}_{\text{total}}(\theta) = \mathcal{L}_{\text{separation}}(\theta) + \mathcal{L}_{\text{diversity}}(\theta)$ ,then at the collapsed solution $\theta^*$ we have

$$\mathcal{L}_{\text{total}}(\theta^*) = \mathcal{L}_{\text{separation}}(\theta^*) + \times 2 \geq 2.$$

Thus *no* collapsed solution can have zero total loss with diversity penalty. Hence, diversity prevents degeneracy.

**Compactness from separation.**

Although the diversity term penalizes an *overly* tight cluster, the separation objective $\mathcal{L}_{\text{separation}}(\theta)$ still rewards each class-$k$ point $z_i = f_\theta(x_i)$ for lying near its center $c_k$. If a point drifts arbitrarily far from $c_k$, $\mathcal{L}_{\text{separation}}$ grows, hurting the total objective. Therefore, at the optimum $\theta^*$, each class remains *relatively compact* around $c_k$ (to keep the separation cost low), yet not collapsed to a single point (to keep diversity cost low).

**Conclusion.** By combining separation and diversity:

- *Non-collapse:* The diversity penalty ensures that no class fully collapses onto one coordinate.

- *Compactness:* The separation objective still keeps each $z_i^*$ near its respective $c_{y_i}$, so all points of class $k$ form a cluster around $c_k$ with moderate scatter.

This balance yields robust representations that preserve in-class variability *and* clear separation from other classes (Goyal et al., 2020).

**Conclusion**

By introducing a *diversity loss*, we ensure that any attempt to collapse all class members to a single point incurs a large penalty, thus preventing the degenerate mappings that minimize the separation loss alone. This mechanism encourages the learned representations to maintain a more realistic spread of samples within each class, ultimately improving *generalization* and *robustness* in anomaly detection tasks. In Appendix D, we compare WSAD-DT with and without the diversity term, providing empirical evidence that the inclusion of the diversity term leads to improved anomaly detection.

## F. Addtional AUC-PR results

In addition to the AUC-ROC results presented in the main paper, we also report the **AUC-PR** (Area Under the Precision-Recall curve) (Davis & Goadrich, 2006) scores for all competing methods. Table 5 summarizes these additional experiments on the same benchmark datasets described in Section H.

- **Consistency with AUC-ROC.** As with ROC-AUC, WSAD-DT consistently achieves strong AUC-PR performance across the majority of datasets. This reaffirms its robustness under heavily imbalanced scenarios, where AUC-PR is often considered more informative than AUC-ROC.

- **Robust Ranking.** Based on WSAD-DT's average rank in AUC-PR, we conclude that it not only effectively separates anomalies from normal points but also preserves high precision under severe class imbalance, where the prevalence of anomalies is especially low.

Overall, the AUC-PR results are consistent with the main findings based on AUC-ROC, further reinforcing the effectiveness of WSAD-DT under weak supervision.

*Table 5.* This table presents the results of all algorithms using the default parameters outlined in the original paper in the metric AUC-PR.

| Dataset | WSAD-DT | DeepSAD | DevNet | FeaWAD | GANAnomaly | PreNet | ROSAS | XGBOD |
|---|---|---|---|---|---|---|---|---|
| Optdigits | 0.9890 (3) | 0.9469 (5) | 0.9830 (4) | 0.8320 (6) | 0.0376 (8) | 0.9991 (2) | **0.9994 (1)** | 0.7631 (7) |
| Lymphography | **1.0000 (1)** | **1.0000 (1)** | 0.9444 (4) | 0.7278 (6) | 0.5051 (7) | **1.0000 (1)** | * | 0.8528 (5) |
| Pendigits | 0.9915 (2) | 0.9265 (6) | 0.9313 (5) | 0.5205 (7) | 0.2064 (8) | 0.9454 (3) | **0.9946 (1)** | 0.9392 (4) |
| Vertebral | **0.6342 (1)** | 0.4025 (4) | 0.1538 (7) | 0.3150 (6) | 0.1136 (8) | 0.6242 (2) | 0.5161 (3) | 0.3958 (5) |
| Wdbc | **1.0000 (1)** | 0.9722 (3) | **1.0000 (1)** | 0.9500 (4) | 0.7641 (8) | 0.9306 (6) | 0.8718 (7) | 0.9444 (5) |
| Wpbc | 0.4495 (2) | 0.4121 (3) | 0.3424 (5) | 0.3296 (6) | 0.2606 (7) | **0.4858 (1)** | * | 0.3744 (4) |
| Stamps | **0.9254 (1)** | 0.7155 (4) | 0.5349 (6) | 0.5357 (5) | 0.2316 (8) | 0.8084 (2) | 0.7703 (3) | 0.5232 (7) |
| Satimage-2 | 0.9122 (2) | 0.8694 (6) | 0.8624 (7) | 0.8701 (5) | 0.3463 (8) | 0.8711 (4) | 0.9004 (3) | **0.9163 (1)** |
| Spambase | 0.9283 (2) | 0.8650 (5) | 0.9005 (3) | * | 0.4589 (7) | 0.8550 (6) | 0.9004 (4) | **0.9451 (1)** |
| Thyroid | **0.8894 (1)** | 0.7866 (5) | 0.7847 (6) | 0.6200 (7) | 0.2137 (8) | 0.8263 (4) | 0.8878 (2) | 0.8465 (3) |
| Mnist | 0.9040 (2) | 0.8078 (5) | 0.7978 (6) | 0.4794 (7) | 0.2665 (8) | 0.8492 (4) | 0.8849 (3) | **0.9114 (1)** |
| Yeast | 0.4716 (4) | 0.4807 (2) | 0.3920 (6) | 0.3472 (7) | 0.3239 (8) | 0.4780 (3) | **0.4888 (1)** | 0.4262 (5) |
| Cardio | 0.9450 (2) | **0.9465 (1)** | 0.8693 (5) | 0.5509 (8) | 0.5967 (7) | 0.8629 (6) | 0.8716 (4) | 0.9265 (3) |
| Vowels | 0.8127 (3) | **0.8205 (1)** | 0.4429 (6) | 0.2680 (7) | 0.2448 (8) | 0.7695 (5) | 0.8090 (4) | 0.8154 (2) |
| Wine | **1.000 (1)** | **1.0000 (1)** | 0.9639 (5) | 0.8905 (6) | 0.2366 (7) | **1.0000 (1)** | * | **1.0000 (1)** |
| Magic.gamma | 0.8767 (2) | 0.8449 (4) | 0.6975 (6) | 0.6480 (7) | 0.5577 (8) | 0.8083 (5) | 0.8718 (3) | **0.8854 (1)** |
| Ionosphere | 0.9658 (2) | 0.9541 (3) | 0.5213 (7) | 0.4288 (8) | 0.6709 (6) | 0.7566 (5) | 0.7942 (4) | **0.9659 (1)** |
| Glass | **0.8310 (1)** | 0.5999 (4) | 0.1867 (6) | 0.1286 (8) | 0.1365 (7) | 0.6732 (3) | 0.4341 (5) | 0.7455 (2) |
| Breastw | 0.9764 (2) | 0.9233 (6) | **0.9895 (1)** | * | 0.9261 (5) | 0.9349 (4) | 0.8082 (7) | 0.9725 (3) |
| Yelp | **0.3663 (1)** | 0.2899 (3) | 0.1321 (6) | 0.0703 (8) | 0.0817 (7) | 0.2564 (4) | 0.3523 (2) | 0.1896 (5) |
| Imdb | **0.2335 (1)** | 0.1602 (2) | 0.0752 (6) | 0.0645 (7) | 0.0499 (8) | 0.1434 (4) | 0.1541 (3) | 0.1203 (5) |
| MNIST-C-Fog | **1.0000 (1)** | **1.0000 (1)** | **1.0000 (1)** | 0.8680 (7) | 0.1794 (8) | **1.0000 (1)** | **1.0000 (1)** | 0.9960 (6) |
| MNIST-C-canny-Edges | **0.9974 (1)** | 0.9655 (5) | 0.9626 (6) | 0.6155 (7) | 0.0832 (8) | 0.9860 (3) | 0.9952 (2) | 0.9741 (4) |
| MVTec-AD-Zipper | **0.8718 (1)** | 0.8021 (3) | 0.6757 (6) | 0.3894 (8) | 0.6298 (7) | 0.6901 (5) | 0.8557 (2) | 0.7750 (4) |
| MNIST-C-Stripe | **1.0000 (1)** | **1.0000 (1)** | **1.0000 (1)** | 0.9421 (7) | 0.6279 (8) | **1.0000 (1)** | **1.0000 (1)** | 0.9992 (6) |
| Skin | **0.9990 (1)** | 0.9972 (3) | 0.9446 (5) | * | 0.2519 (6) | † | 0.9902 (4) | 0.9979 (2) |
| Fraud | 0.7437 (2) | 0.7361 (3) | 0.4941 (6) | 0.2695 (7) | 0.6274 (5) | † | 0.6832 (4) | **0.7978 (1)** |
| Http | **0.9985 (1)** | 0.9947 (2) | 0.9908 (4) | * | 0.2886 (6) | † | 0.9798 (5) | 0.9921 (3) |
| Cover | 0.9647 (4) | 0.9851 (2) | 0.9512 (5) | 0.1392 (6) | 0.0091 (7) | † | **0.9861 (1)** | 0.9810 (3) |
| Shuttle | 0.9901 (2) | 0.9865 (3) | 0.9680 (5) | 0.9546 (6) | 0.9359 (7) | † | 0.9773 (4) | **0.9978 (1)** |
| AVGRank | 1.70 | 3.23 | 4.90 | 6.77 | 7.27 | 4.10 | 3.60 | 3.37 |
| p-value | N/A | 0.00192370 (+) | 0.00016651 (+) | 0.00000005 (+) | 0.00000005 (+) | 0.00044749 (+) | 0.00262462 (+) | 0.04616544 (+) |

Values marked with † indicate that no result was available within 12 hours.
Values marked with * indicate that a runtime error occurred during execution.

# G. Algorithm details

In Algo. 1 we describe WSAD-DT in detail. The core of WSAD-DT is to leverage both labeled anomalies and unlabeled data in a weakly supervised setting by training multiple models (an ensemble) and then aggregating their anomaly scores. First, the unlabeled dataset is split into $M$ disjoint subsets; each subset is augmented with the same small set of labeled anomalies, ensuring that every model has consistent information about the known anomalies while observing different portions of unlabeled data. This yields $M$ distinct training sets, each used to train a separate neural network mapping inputs to a latent representation. Within the latent space, two distinct centers are maintained, one for normal points ($c_0$) and another for anomalies ($c_1$). A dual-kernel approach drives the representations toward the correct center and away from the incorrect one. Specifically, a light-tailed kernel is applied to in-class distances, causing normal or anomalous samples to cluster tightly around their correct center, whereas a heavy-tailed kernel is applied to out-of-class distances, keeping mismatched samples at a broader margin. This two-part strategy facilitates both tight grouping for in-class points and clear separation between classes, which a single kernel cannot achieve simultaneously. A crucial challenge is the risk of degenerate solutions where all points of a given class collapse onto a single coordinate in the latent space. To mitigate this, WSAD-DT introduces a diversity regularization term that penalizes pairs of points in the same class that are mapped too closely together. This penalty protects intra-class variability by discouraging trivial collapse onto each center. Finally, once every network in the ensemble is trained, each model's anomaly scores for a new sample are combined, by averaging, to yield a final ensemble score. This last step enhances stability and robustness, especially given the scant availability of labeled anomalies, by capitalizing on

diverse perspectives of the unlabeled data while retaining consistent guidance from the same known anomalies.

---

**Algorithm 1** WSAD-DT

---

**Require:** *Labeled anomalies $\mathcal{D}_L = \{(x_i, 1)\}$, Unlabeled data $\mathcal{D}_U = \{(x_j, 0)\}$, Ensemble size $M$, mini-batch size $b$, max epoch maxEpoch*

1: **Partition** $\mathcal{D}_U$ into $M$ disjoint subsets $\{\mathcal{D}_U^{(m)}\}_{m=1}^M$, each combined with $\mathcal{D}_L$ to form $\mathcal{D}^{(m)}$.

2: **for** $m = 1$ to $M$ **do**

3:     Initialize network parameters $\Theta_m$ and centers $c_0$, $c_1$.

4:     **for epoch** $= 1$ to *maxEpoch* **do**

5:       **for each** mini-batch $\mathcal{W}' \subset \mathcal{D}^{(m)}$ (of size $b$) **do**

6:         **Step 1. Compute embeddings** $z_i = f_{\Theta_m}(x_i)$ for all $x_i \in \mathcal{W}'$.

7:         **Step 2. Compute separation loss** ($\mathcal{L}_{\text{seperation}}$):

$$\mathcal{L}_{\text{separation}}(\theta) = \underbrace{\sum_{(x,y) \in \mathcal{W}'} \ell_{\text{separation}}(\theta; x, y)}_{\text{sum of single-sample losses}}$$

8:         **Step 3. Subsampling for diversity:**

9:         Let $\mathcal{W}'_0 = \{z_i \mid (x_i, 0) \in \mathcal{W}'\}$ and $\mathcal{W}'_1 = \{z_i \mid (x_i, 1) \in \mathcal{W}'\}$.

10:        **Uniformly randomly sample** $b_s = \lfloor \sqrt{b} \rfloor$ points from each $\mathcal{W}'_k$, yielding subsets $\tilde{\mathcal{W}}'_k \subseteq \mathcal{W}'_k$.

11:        **Step 4. Compute diversity loss** ($\mathcal{L}_{\text{diversity}}$):

$$\mathcal{L}_{\text{diversity}}(\theta) = \sum_{k \in \{0,1\}} \frac{1}{|\tilde{\mathcal{W}}'_k|^2} \sum_{z_i, z_j \in \tilde{\mathcal{W}}'_k} \exp\left(-\frac{\|z_i - z_j\|}{\sigma^2_{\tilde{\mathcal{W}}'_k}}\right).$$

12:        **Step 5. Gradient update**:

$$\Theta_m \leftarrow \Theta_m - \eta \cdot \nabla_{\Theta_m}\left(\mathcal{L}_{\text{separation}}(\theta) + \mathcal{L}_{\text{diversity}}(\theta)\right).$$

13:       **end for**

14:     **end for**

15: **end for**

16: **Inference**: For each test sample $x$, each model outputs

$$\phi_m(x) = 1 - \frac{\mathcal{K}_{\text{heavy}}(\|f_{\Theta_m}(x) - c_0\|)}{\mathcal{K}_{\text{heavy}}(\|f_{\Theta_m}(x) - c_0\|) + \mathcal{K}_{\text{heavy}}(\|f_{\Theta_m}(x) - c_1\|)}.$$

17: **Aggregate** ensemble outputs:

$$\phi(x) = \frac{1}{M} \sum_{m=1}^M \phi_m(x).$$

    Final anomaly scores $\phi(x)$.

---

# H. Experiment details

**Experiment Setup**

All experiments were conducted on a workstation equipped with an Intel Core i7-10700K CPU (3.8 GHz) and 32 GB of RAM. We repeat all experiments on three different splits and report average results. Performance is measured with AUC-ROC and AUC-PR (Davis & Goadrich, 2006), and significance is determined via a paired Wilcoxon signed-rank test with Holm-Bonferroni correction (McDonald, 2014). All features are scaled to $[0, 1]$ using MinMaxScaler (Pedregosa et al., 2011). For stochastic methods, five random seeds $\{0,1,2,100,1000\}$ are used, and default hyperparameters according to (Xu, 2023).

**Datasets and Splits**

We adopt a diverse set of real-world anomaly detection benchmarks from *AdBench* (Han et al., 2022) (Table 6), each split 70%–30% for training and testing via stratified sampling to preserve anomaly ratios. In line with (Han et al., 2022), only a small fraction of anomalies in the training split is labeled (weak supervision), specifically labeling either 5 anomalies or $p\%$ of anomalies—whichever is greater—to ensure a minimal but consistent supervisory signal across all experiments. These labeled anomalies are selected uniformly at random from the available anomalies. This setup reflects real-world conditions, where annotated anomalies are scarce and often fail to capture the breadth of abnormal behaviors. We further analyze different fractions of labeled anomalies in Appendix N and contamination in the training data in Appendix M.

| Dataset | # Instances | # Dimensions | # Anomalies (%) |
|---|---|---|---|
| Optdigits | 5216 | 64 | 150(0.0288) |
| Lymphography | 148 | 18 | 6(0.0405) |
| Pendigits | 6870 | 16 | 156(0.0227) |
| Vertebral | 240 | 6 | 30(0.1250) |
| Wdbc | 367 | 30 | 10(0.0272) |
| Cardiotocography | 2114 | 21 | 466(0.2204) |
| Wpbc | 198 | 33 | 47(0.2374) |
| Stamps | 340 | 9 | 31(0.0912) |
| Satimage-2 | 5803 | 36 | 71(0.0122) |
| Spambase | 4207 | 57 | 1679(0.3991) |
| Thyroid | 3772 | 6 | 93(0.0247) |
| Mnist | 7603 | 100 | 700(0.0921) |
| Yeast | 1484 | 8 | 507(0.3416) |
| Cardio | 1831 | 21 | 176(0.0961) |
| Vowels | 1456 | 12 | 50(0.0343) |
| Wine | 129 | 13 | 10(0.0775) |
| Magic.gamma | 19020 | 10 | 6688(0.3516) |
| Ionosphere | 351 | 32 | 126(0.3590) |
| Glass | 214 | 7 | 9(0.0421) |
| Breastw | 683 | 9 | 239(0.3499) |
| Yelp | 10000 | 512 | 500(0.0500) |
| Imdb | 10000 | 512 | 500(0.0500) |
| MNIST-C-Fog | 10000 | 512 | 500(0.0500) |
| MNIST-C-canny-Edges | 10000 | 512 | 500(0.0500) |
| MVTec-AD-Zipper | 10000 | 512 | 500(0.0500) |
| MNIST-C-Striper | 10000 | 512 | 500(0.0500) |
| Skin | 245057 | 3 | 50859(0.2075) |
| Http | 567498 | 3 | 2211(0.0039) |
| Cover | 286048 | 10 | 2747(0.0096) |
| Shuttle | 49097 | 9 | 3511(0.0715) |

*Table 6.* Statistics of the used datasets

**Implementation and Code**

Our code is implemented in PyTorch and builds on top of the `DeepOD` and `PyOD` libraries (Zhao et al., 2019; Xu, 2023). Our anonymous code repository: **Link (Anonymous)**.

**Baselines**

We compare our method (**WSAD-DT**) with state-of-the-art anomaly detection baselines, including *DeepSAD*, *DevNet*, *FeaWAD*, *GANomaly*, *PReNet*, *RoSAS*, and ensemble-based *XGBOD*. For each baseline, we used the open-source implementation from (Xu et al., 2023a; Han et al., 2022); further details can be found in Appendix I. By default, XGBOD employs multiple unsupervised detectors (including KNN (Ramaswamy et al., 2000) and LOF (Breunig et al., 2000)). However,

on datasets exceeding 100000 samples, these two detectors took over 12 hours to complete. We therefore omit KNN and LOF in such large-scale cases to keep XGBOD's runtime feasible, while still leveraging the remaining detectors for reliable anomaly scores.

## Kernel Choices & Hyperparameters

One key novelty in **WSAD-DT** is the *dual-kernel* design: a *light-tailed* kernel for in-class similarity and a *heavy-tailed* kernel for out-of-class separation. Below, we outline each kernel and its bandwidth parameters ($\sigma$, $\nu$).

### H.1. Light-Tailed Kernel (Gaussian)

$$\mathcal{K}_{\text{light}}(d_{i,k}) \; = \; \exp\!\Big(\!-\frac{d_{i,k}}{2\,\sigma_k^2}\Big),$$

- $\sigma_{\mathcal{U}} = 0.5$ for the normal center, ensuring a tighter cluster for normal data.

- $\sigma_{\mathcal{A}} = 1.0$ for the anomaly center, allowing a slightly looser cluster for more diverse anomalies.

### H.2. Heavy-Tailed Kernel (Student-$t$)

$$\mathcal{K}_{\text{heavy}}(d_{i,k}) \; = \; \big(1 + \tfrac{d_{i,k}}{\nu}\big)^{-\frac{\nu+1}{2}}.$$

- Default $\nu = 0.2$. A smaller $\nu$ corresponds to a heavier tail, which helps separate out-of-class points more aggressively.

## Diversity Term

A key risk of minimizing only the separation loss is *collapse*, where all points in a class map to the same latent coordinate. To counteract this, we introduce a class-specific *exponential diversity penalty*. For each class $C \in \{\mathcal{A}, \mathcal{U}\}$, we first measure the average pairwise similarity:

$$k(C; \theta) \; = \; \frac{1}{|C|^2} \sum_{i,j \in C} \exp\!\Big(\!-\frac{\|f_\theta(x_i) - f_\theta(x_j)\|}{\sigma_C^2}\Big),$$

where $\sigma_C > 0$ is a fixed bandwidth parameter for class $C$. The overall *diversity loss* is then defined as

$$\mathcal{L}_{\text{diversity}}(\theta) \; = \; k(\mathcal{A}; \theta) \; + \; k(\mathcal{U}; \theta).$$

**Role of $\sigma_C$.** In practice, the parameter $\sigma_C$ controls how strongly the method penalizes points that are closely mapped in latent space:

- **Small** $\sigma_C$ (e.g., $\sigma_{\text{diversity},\mathcal{U}} = 0.1$ for normal data) leads to a weaker penalty for moderate distances, so partial clustering incurs but still effectively prevents any accidental clustering into a single point.

- **Larger** $\sigma_C$ (e.g., $\sigma_{\text{diversity},\mathcal{A}} = 1.0$ for anomalies) allows more dispersion among inherently diverse anomalies, since moderate pairwise distances will incur a sufficient penalty.

A high value of $k(C; \theta)$ indicates that points in class $C$ are excessively close in the latent space, triggering a larger penalty and thus discouraging trivial collapse. By adjusting $\sigma_C$ appropriately for each class, one can balance intra-class variability against the risk of collapse. In practice, these parameters may be selected via cross-validation or a small parameter search, depending on dataset size and diversity.

A detailed ablation study on hyperparameter selection is provided in Appendix O, offering comprehensive insights into the impact of each parameter on model performance.

## Ensemble Setup ($M$ splits)

By default, **WSAD-DT** uses $M = 5$ ensemble splits (unless otherwise noted). An ablation over $M \in \{1, 3, 5\}$ is reported in the Appendix L, showing that $M = 5$ typically yields the best or near-best performance.

**Evaluation Metrics & Statistical Significance**

**Metrics**

We use two widely adopted metrics to evaluate anomaly detection models:

1. **ROC-AUC (Receiver Operating Characteristic—Area Under the Curve)** (Davis & Goadrich, 2006): This measures the trade-off between the *True Positive Rate (TPR)* and the *False Positive Rate (FPR)* over different decision thresholds. A higher ROC-AUC indicates stronger discriminative power in distinguishing anomalies from normal points.

2. **PR-AUC (Precision–Recall—Area Under the Curve)** (Davis & Goadrich, 2006): Unlike ROC-AUC, PR-AUC focuses on *Precision* (the fraction of detected anomalies that are actually anomalous) and *Recall* (the fraction of anomalies correctly identified).

By reporting both metrics, we gain a more comprehensive view of each method's effectiveness in detecting anomalies across varying degrees of class imbalance.

**Statistical Testing**

We adopt the paired Wilcoxon signed-rank test with Holm–Bonferroni correction to compare **WSAD-DT** against each baseline at $\alpha = 0.05$. We highlight "+" if **WSAD-DT** is significantly superior, "−" if worse, and "≈" otherwise. In the reporting tables, we list the adjusted p-values with Holm–Bonferroni correction.

**Ranking**

All tables include an "average rank" across all datasets; a smaller rank value indicates better overall performance.

# I. Implementation Details

All algorithms are implemented as described in their respective original papers and integrated into the publicly available *DeepOD* library (Xu, 2023), following a uniform training protocol to ensure fair comparisons across methods. Each experiment is conducted on the same data splits, using consistent data loading and preprocessing procedures.

**Network Architecture.** We employ a three-layer feed-forward encoder with hidden sizes of 100 and 50, followed by a projection into a final embedding dimension of 128, aligning with configurations used in approaches such as *DeepSAD*. Formally, the neural network has the layers:

$$\text{input dimension} \ \rightarrow \ 100 \ \rightarrow \ 50 \ \rightarrow \ 128.$$

**Activation Functions.** In our proposed method WSAD-DT, we utilize the Scaled Exponential Linear Unit (SELU) activation function at each layer, which we found to stabilize training and often accelerate convergence. For other baselines, we adhere to the recommended activation functions specified in their original publications, such as ReLU or LeakyReLU. SELU activation function is defined as:

$$\text{SELU}(x) = \begin{cases} \lambda x & \text{if } x > 0, \\ \lambda \alpha (e^x - 1) & \text{if } x \leq 0, \end{cases}$$

where:

- $\lambda \approx 1.0507$ is the scaling parameter,

- $\alpha \approx 1.67326$ is the parameter that controls the slope for negative inputs.

**Optimization.** All models are trained for 100 epochs using the Adam optimizer with a learning rate of $1 \times 10^{-3}$ and a weight decay of $1 \times 10^{-5}$. We use the standard Adam hyperparameters ($\beta_1 = 0.9$, $\beta_2 = 0.999$). Batches of size 64 are used for each training step (Table 7).

**Remark:** Importantly, apart from using the SELU activation function, we did not implement any other modifications. To ensure the comparison is as fair as possible to the comparison method, our method and the underlying network architecture, along with the corresponding hyperparameters, remain the same as recommended by (Xu, 2023).

| Parameter | Value |
|---|---|
| **GENERAL TRAINING** | |
| Batch size | 64 |
| Learning rate | 1e–3 |
| Epochs | 100 |
| **ARCHITECTURE** | |
| Feed Forward | InputDim-100-50-embedding-dim |
| Embedding size | 128 |
| Activation function | SELU |
| **OPTIMIZER** | |
| Optimizer | Adam |
| Momentum $\beta_1$ | 0.9 |
| Momentum $\beta_2$ | 0.999 |
| Weight decay | 1e–5 |

*Table 7.* Neuralnetwork and training setting of WSAD-DT

## J. Scalability Test

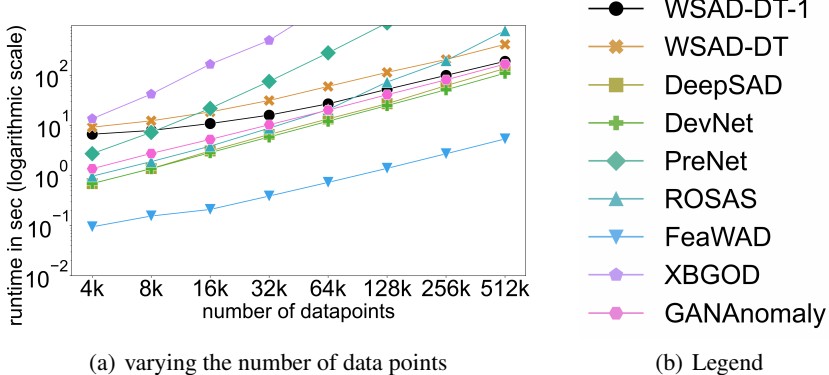

(a) varying the number of data points  (b) Legend

*Figure 2.* Analyzing the runtime performance for WSAD-DT and the competitors.

To assess the scalability of our proposed method (WSAD-DT), we measure its runtime performance on an increasing number of data points and compare it against several baselines. Figure 2 illustrates the average wall-clock time (in seconds) as a function of dataset size for WSAD-DT and other competing methods.

**Experimental Setup.** We simulate large-scale synthetic datasets by primarily drawing normal samples from a 16-dimensional Gaussian and injecting a small fraction (5%-10%) of anomalies sampled from a distinct distribution. We label these injected points as anomalies and combine them with the normal samples. This yields a controlled dataset where the fraction of anomalies remains fixed while we grow the overall dataset size from 4,000 samples up to 512,000 samples. Each size is run three times with different random seeds, and we report the average wall-clock time for each approach. For this experiment, we used the default setup described in Appendix I, with the exception that we fixed the number of epochs to 10 for all algorithms in this experiment.

- **Asymptotic Behavior.** As illustrated in Figure 2(a), WSAD-DT exhibits runtime scaling similar to DeepSAD and other deep anomaly detection approaches as the number of samples increases. This behavior is expected in standard mini-batch-based deep learning pipelines. Although WSAD-DT includes a diversity penalty term, its computational overhead remains modest. Specifically, the diversity penalty is evaluated only on each mini-batch and further reduced by subsampling pairs of points, making its contribution to overall runtime negligible in practice. In contrast, the

ensemble method XGBOD (which uses KNN (Ramaswamy et al., 2000) and LOF (Breunig et al., 2000) as base estimators) experiences higher computational overhead on large datasets, consistent with our observations. Meanwhile, WSAD-DT maintains a relatively consistent runtime profile, particularly with mini-batch optimization.

- **Impact of Ensemble Size.** In Fig. 2(b), WSAD-DT-1 refers to WSAD-DT with a single ensemble, whereas WSAD-DT represents the default setting with five ensembles. The runtime does increase proportionally with the number of ensemble splits in WSAD-DT since each component model is trained on a distinct partition of unlabeled data. However, we set the default number of ensembles to a moderate value (e.g., $M = 5$), striking a practical balance between computational overhead and performance gains.

Overall, WSAD-DT demonstrates competitive runtime scalability, making it well-suited for large-scale anomaly detection tasks where both robust performance and manageable computational cost are required.

## K. Ablation study of light-tailed, heavy-tailed, and dual-tailed kernel

We conducted an ablation study by separately evaluating the light-tailed kernel, the heavy-tailed kernel, and their combination (dual-tailed kernel). We set the number of ensembles to 1 for all different configurations (light-tailed kernel, heavy-tailed kernel, dual-tailed kernel). Our findings (Table 8) reveal that while each kernel on its own provides a decent level of performance, combining both kernels clearly improves the model's ability to distinguish anomalies from normal points. Specifically, the dual-tailed kernel configuration yields higher AUC-ROC scores compared to either kernel used in isolation. This result underscores the complementary benefits of the two kernels, with the light-tailed kernel component enforcing tight clustering for in-class points and the heavy-tailed component preserving greater separation for out-of-class samples.

*Table 8.* We present the AUC-ROC values for various configurations in WSAD-DT. Specifically, we compare the AUC-ROC performance across three settings: single kernels (light-tailed kernel and heavy-tailed kernel) and dual-tailed kernel.

| Dataset | WSAD-DT | Light-tailed kernel | Heavy-tailed kernel |
|---|---|---|---|
| Optdigits | **0.9984 (1)** | 0.8875 (3) | 0.9845 (2) |
| Lymphography | **0.9961 (1)** | 0.9787 (2) | 0.9593 (3) |
| Pendigits | **0.9997 (1)** | 0.9477 (3) | 0.9991 (2) |
| Vertebral | **0.8827 (1)** | 0.8354 (2) | 0.7778 (3) |
| Wdbc | **1.0000 (1)** | 0.9979 (3) | 0.9995 (2) |
| Cardiotocography | **0.9347 (1)** | 0.8899 (2) | 0.8894 (3) |
| Wpbc | **0.6801 (1)** | 0.6724 (2) | 0.5807 (3) |
| Stamps | **0.9841 (1)** | 0.8769 (3) | 0.9456 (2) |
| Satimage-2 | **0.9954 (1)** | 0.9811 (3) | 0.9941 (2) |
| Spambase | **0.9317 (1)** | 0.8842 (3) | 0.9202 (2) |
| Thyroid | **0.9911 (1)** | 0.9667 (3) | 0.9889 (2) |
| Mnist | **0.9820 (1)** | 0.8293 (3) | 0.9696 (2) |
| Yeast | **0.6566 (1)** | 0.6409 (2) | 0.5854 (3) |
| Cardio | **0.9908 (1)** | 0.9505 (3) | 0.9741 (2) |
| Vowels | **0.9824 (1)** | 0.8737 (3) | 0.9629 (2) |
| Wine | **1.0000 (1)** | 0.9923 (2) | 0.9769 (3) |
| Magic.gamma | **0.8974 (1)** | 0.8375 (3) | 0.8649 (2) |
| Ionosphere | **0.9836 (1)** | 0.9510 (3) | 0.9651 (2) |
| Glass | 0.9794 (2) | 0.9543 (3) | **0.9964 (1)** |
| Breastw | 0.9828 (2) | 0.9658 (3) | **0.9844 (1)** |
| Yelp | **0.7811 (1)** | 0.7355 (2) | 0.7189 (3) |
| Imdb | **0.6882 (1)** | 0.6074 (3) | 0.6228 (2) |
| MNIST-C-Fog | **1.0000 (1)** | **1.0000 (1)** | 0.9998 (3) |
| MNIST-C-Canny-Edges | **0.9999 (1)** | 0.9846 (3) | 0.9989 (2) |
| MVTec-AD-Zipper | **0.9280 (1)** | 0.8468 (3) | 0.9140 (2) |
| MNIST-C-Stripe | **1.0000 (1)** | **1.0000 (1)** | **1.0000 (1)** |
| Skin | **0.9998 (1)** | 0.9991 (2) | 0.9987 (3) |
| Fraud | **0.9485 (1)** | 0.9346 (3) | 0.9416 (2) |
| Http | **1.0000 (1)** | **1.0000 (1)** | **1.0000 (1)** |
| Cover | **0.9995 (1)** | 0.9896 (3) | 0.9973 (2) |
| Shuttle | 0.9935 (2) | 0.9862 (3) | **0.9961 (1)** |
| AvgRank | 1.10 | 2.55 | 2.13 |
| p-values | NA | 0.00001200 (+) | 0.00015176 (+) |

# L. Ablation study on the effect of different numbers of ensembles

Below, we present an ablation study evaluating the effect of varying the number of ensemble models (#ensembles) in WSAD-DT from 1 to 3 to 5. Table 9 reports AUC-ROC scores for a selection of benchmark datasets. A single model (#1) already achieves strong results, yet using #3 ensembles yields a clear boost on most datasets. Increasing the ensemble size to #5 generally delivers the highest overall performance.

**Key Observations**

1. **#1 vs. #3:** Introducing ensembles (i.e., going from 1 to 3) consistently improves robustness. The models benefit from diverse *views* of the unlabeled data, while all leveraging the same limited set of labeled anomalies.

2. **#3 vs. #5:** Although gains can be smaller after #3, using #5 often still improves AUC-ROC, suggesting further diversity helps in certain datasets with complex structures or highly imbalanced anomalies.

3. **Computational Cost vs. Accuracy:** While larger ensembles typically perform better, they also introduce additional training overhead. In practice, #3 or #5 often provide a good balance between computational cost and accuracy.

Overall, the results confirm that using an ensemble of models, each trained on a distinct partition of the unlabeled data while sharing the same few labeled anomalies, significantly enhances performance. In practice, #5 ensembles are often a good default, balancing the improved robustness from ensemble diversity against the computational cost.

*Table 9.* This table presents the AUC-ROC results of WSAD-DT under different number of ensembles ($M$).

| Dataset | #5-Spilt | #1-Spilt | #3–Spilt |
|---|---|---|---|
| Optdigits | **0.9996 (1)** | 0.9984 (3) | 0.9995 (2) |
| Lymphography | **1.0000 (1)** | 0.9961 (3) | **1.0000 (1)** |
| Pendigits | **0.9998 (1)** | 0.9997 (2) | 0.9997 (2) |
| Vertebral | **0.9051 (1)** | 0.8827 (3) | 0.8939 (2) |
| Wdbc | **1.0000 (1)** | **1.0000 (1)** | 0.9990 (3) |
| Wpbc | 0.6915 (2) | 0.6801 (3) | **0.6918 (1)** |
| Stamps | **0.9902 (1)** | 0.9841 (3) | 0.9883 (2) |
| Satimage-2 | **0.9973 (1)** | 0.9954 (3) | 0.9962 (2) |
| Spambase | **0.9499 (1)** | 0.9317 (3) | 0.9454 (2) |
| Thyroid | **0.9960 (1)** | 0.9911 (3) | 0.9951 (2) |
| Mnist | **0.9883 (1)** | 0.9820 (3) | 0.9867 (2) |
| Yeast | 0.6545 (2) | **0.6566 (1)** | 0.6512 (3) |
| Cardio | 0.9908 (2) | 0.9908 (2) | **0.9919 (1)** |
| Vowels | 0.9736 (3) | **0.9824 (1)** | 0.9778 (2) |
| Wine | **1.0000 (1)** | **1.0000 (1)** | **1.0000 (1)** |
| Magic.gamma | **0.9166 (1)** | 0.8974 (3) | 0.9150 (2) |
| Ionosphere | 0.9779 (3) | **0.9836 (1)** | 0.9792 (2) |
| Glass | **0.9830 (1)** | 0.9794 (2) | 0.9794 (2) |
| Breastw | **0.9894 (1)** | 0.9828 (3) | 0.9883 (2) |
| Yelp | **0.8311 (1)** | 0.7811 (3) | 0.8008 (2) |
| Imdb | **0.7623 (1)** | 0.6882 (3) | 0.7379 (2) |
| MNIST-C-Fog | **1.0000 (1)** | **1.0000 (1)** | **1.0000 (1)** |
| MNIST-C-canny-Edges | **0.9999 (1)** | **0.9999 (1)** | **0.9999 (1)** |
| MVTec-AD-Zipper | 0.9307 (2) | 0.9280 (3) | **0.9354 (1)** |
| MNIST-C-Stripe | **1.0000 (1)** | **1.0000 (1)** | **1.0000 (1)** |
| Skin | **0.9998 (1)** | **0.9998 (1)** | **0.9998 (1)** |
| Fraud | **0.9574 (1)** | 0.9485 (3) | 0.9556 (2) |
| Http | **1.0000 (1)** | **1.0000 (1)** | **1.0000 (1)** |
| Cover | 0.9996 (2) | 0.9995 (3) | **0.9997 (1)** |
| Shuttle | **0.9956 (1)** | 0.9935 (2) | 0.9934 (3) |
| AVGRank | 1.30 | 2.20 | 1.73 |
| p-value | NA | 0.00645607 (+) | 0.00947055 (+) |

## M. Analyzing the performance under contamination in the training data

In real-world scenarios, a portion of the unlabeled dataset may itself be contaminated with anomalies, making the weakly supervised setting even more challenging.

**Contamination Procedure.**    We begin by using only $5\%$ of the total anomalies for weakly supervised training, *chosen uniformly at random*, leaving the remaining anomalies "unused." We then define a contamination fraction $p\%$ (relative to these unused anomalies) to artificially degrade the unlabeled set. Specifically, we inject $p\%$ of the leftover anomalies into the unlabeled pool, *mislabeled* as normal points. At the same time, we select the same number of normal points from the unlabeled set and *mislabeled* them as anomalies. This symmetrical label swap increases both label noise and anomaly contamination in the unlabeled data, enabling a systematic evaluation of WSAD-DT under two configurations: WSAD-DT (ensemble size of 5) and WSAD-DT-1 (ensemble size of 1) across various levels of mislabeling and adversarial injection (see Fig. 3i). Using this setup, we run multiple experiments with $p\%$ ranging from $1\%$ (small contamination) to $100\%$ (maximum contamination). Figure 3 shows the AUC-ROC curves of WSAD-DT and several reference methods over eight

representative datasets as contamination increases. Across all levels of contamination, WSAD-DT (blue curve) demonstrates consistently higher or comparable AUC-ROC values compared to other approaches. WSAD-DT-1 with a single ensemble also consistently achieves higher or comparable AUC-ROC values than other approaches, underscoring the advantages of the dual-tailed kernel and kernel-based regularization. Notably, its performance degrades more gracefully than baselines when the contamination rate exceeds 10%. These findings underline two key advantages of WSAD-DT:

1. **Robustness to Noisy Unlabeled Data.** By maintaining dual-tailed separation, WSAD-DT can still isolate anomalies even when contamination is present in the data.

2. **Effective Use of Sparse Labels.** Sharing a small number of labeled anomalies across ensemble splits remains advantageous. Even at higher contamination levels, each component of the ensemble learns to down-weight suspicious unlabeled instances using the limited—but high-value—labeled examples.

Overall, WSAD-DT's resilience to contamination makes it well-suited for practical applications where unlabeled data may not be entirely clean, reinforcing the benefits of a dual-tailed kernel formulation and ensemble-based training under weak supervision.

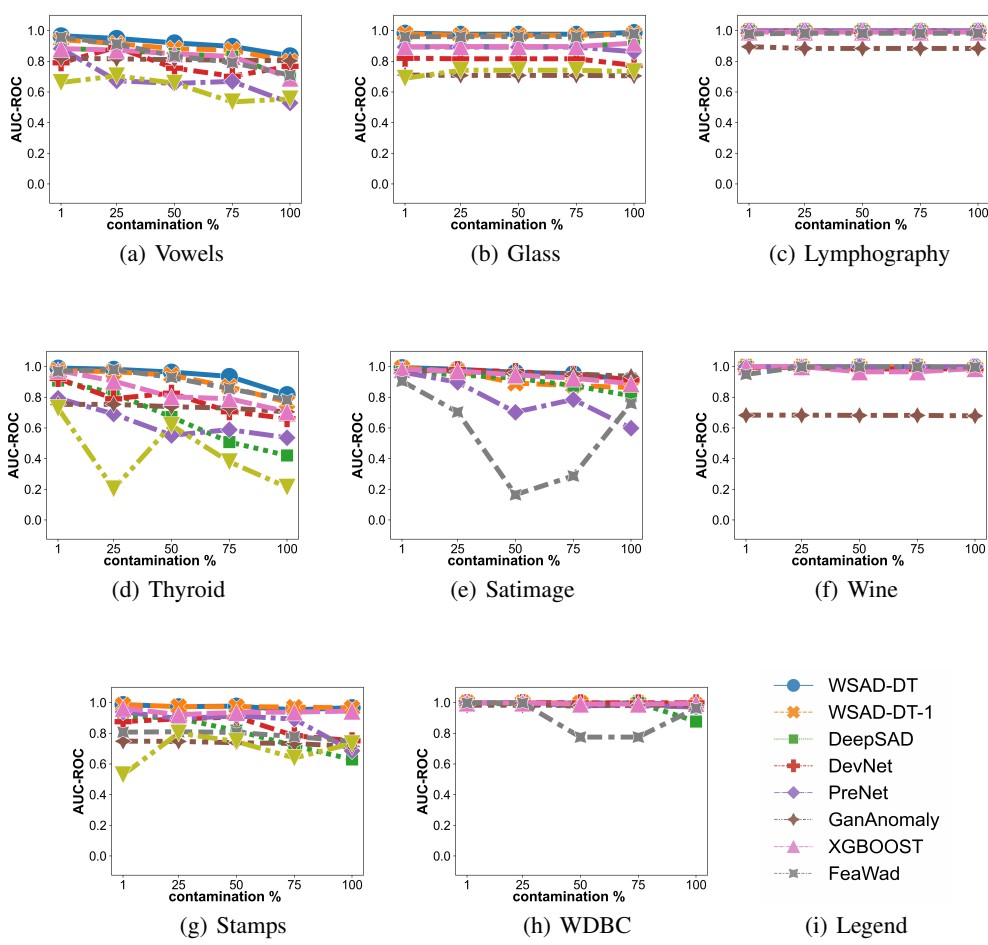

*Figure 3.* Analyzing the performance under contamination in the training data.

# N. Ablation study for different numbers of labeled anomalies

Below, we examine how varying the fraction of labeled anomalies in the training set impacts performance under weak supervision. While our main experiments generally use $5\%$ labeled anomalies, real-world anomaly detection tasks may have even fewer labeled anomalies, or—though still "weakly" supervised—potentially a higher fraction of anomaly labels.

## N.1. Experimental Setup

We select multiple benchmark datasets from our main evaluation and train WSAD-DT using three different proportions of labeled anomalies in the training data: $1\%$, $5\%$, and $10\%$. All other settings (network architecture, batch size, mini-batch sampling, etc.) remain identical to the primary experiments in Section H. Specifically,

- **1% labeled anomalies:** Extremely limited supervision, approaching the scarcity seen in purely unsupervised settings.

- **5% labeled anomalies:** Our default setting is a moderately weakly supervised scenario.

- **10% labeled anomalies:** We have more labeled anomalies, but still lack enough fully supervised data to provide better coverage of anomaly types.

We compare WSAD-DT's performance to other state-of-the-art methods (DeepSAD, DevNet, *etc.*) under these varying amounts of labeled anomalies.

## N.2. Results and Observations

Tables 10 and 11 show the AUC-ROC results for two representative levels: $1\%$ and $10\%$ labeled anomalies. (Our main paper reports the case of $5\%$.)

**Key Observations:**

- **Sensitivity to Labeled Fraction.** As expected, all methods tend to improve with more labeled anomalies, but WSAD-DT consistently outperforms or matches the best baseline at each fraction. Even at *1%* labeled anomalies, it achieves top performance on a majority of datasets.

- **Strong Gains from Limited Labels.** Moving from 1% to 10% labeled anomalies often yield a large jump for WSAD-DT. This suggests that even a modest increase in anomaly labels can substantially reduce false positives and sharpen the separation boundary, especially given the two-center and the dual-tailed kernel approach.

**Conclusion.** WSAD-DT exhibits robust performance across a broad range of labeled anomaly percentages. While having more labeled anomalies generally leads to more accurate and stable decision boundaries, our method consistently maintains a strong lead (or near-lead) even when extremely few (1%) labeled anomalies are available. This resiliency underscores the effectiveness of the dual-kernel framework and ensemble strategy in leveraging limited supervision for improved anomaly detection.

*Table 10.* This table presents the results of all algorithms using the default parameters outlined in the original paper and **1%** of labeled anomalies.

| Dataset | WSAD-DT | DeepSAD | DevNet | FeaWAD | GANAnomaly | PreNet | ROSAS | XGBOD |
|---|---|---|---|---|---|---|---|---|
| Optdigits | 0.9996 (3) | 0.9773 (7) | 0.9959 (4) | 0.9839 (5) | 0.6041 (8) | **1.0000 (1)** | **1.0000 (1)** | 0.9817 (6) |
| Lymphography | **1.0000 (1)** | **1.0000 (1)** | 0.9961 (4) | 0.9787 (6) | 0.8953 (7) | **1.0000 (1)** | * | 0.9903 (5) |
| Pendigits | **0.9998 (1)** | 0.9768 (5) | 0.9682 (6) | 0.7798 (8) | 0.8361 (7) | 0.9832 (4) | 0.9997 (2) | 0.9981 (3) |
| Vertebral | **0.9051 (1)** | 0.7810 (5) | 0.4974 (7) | 0.6399 (6) | 0.3618 (8) | 0.9036 (2) | 0.8377 (3) | 0.8125 (4) |
| Wdbc | **1.0000 (1)** | 0.9990 (3) | **1.0000 (1)** | 0.9979 (4) | 0.9830 (8) | 0.9969 (6) | 0.9933 (7) | 0.9979 (4) |
| Wpbc | **0.6915 (1)** | 0.6144 (4) | 0.6403 (3) | 0.5619 (6) | 0.4635 (7) | 0.6649 (2) | * | 0.6135 (5) |
| Stamps | **0.9902 (1)** | 0.9492 (4) | 0.8909 (6) | 0.8664 (7) | 0.7529 (8) | 0.9697 (2) | 0.9544 (3) | 0.9011 (5) |
| Satimage-2 | **0.9973 (1)** | 0.9715 (5) | 0.9562 (6) | 0.9753 (4) | 0.9783 (3) | 0.9243 (8) | 0.9526 (7) | 0.9866 (2) |
| Spambase | 0.8900 (3) | 0.7591 (5) | 0.9214 (2) | * | 0.6103 (7) | 0.7334 (6) | 0.7933 (4) | **0.9263 (1)** |
| Thyroid | 0.9960 (2) | 0.9782 (4) | 0.9667 (5) | 0.8525 (7) | 0.7795 (8) | 0.9420 (6) | **0.9961 (1)** | 0.9795 (3) |
| Mnist | **0.9767 (1)** | 0.8268 (5) | 0.7492 (7) | 0.6498 (8) | 0.8281 (4) | 0.7907 (6) | 0.8725 (3) | 0.9759 (2) |
| Yeast | 0.6024 (3) | 0.5351 (7) | 0.6081 (2) | **0.6268 (1)** | 0.4884 (8) | 0.5453 (6) | 0.5678 (5) | 0.5797 (4) |
| Cardio | **0.9916 (1)** | 0.9796 (3) | 0.9686 (4) | 0.8559 (8) | 0.9095 (6) | 0.9052 (7) | 0.9672 (5) | 0.9837 (2) |
| Vowels | **0.9736 (1)** | 0.9712 (2) | 0.8839 (6) | 0.7696 (8) | 0.8231 (7) | 0.9300 (5) | 0.9653 (3) | 0.9516 (4) |
| Wine | **1.0000 (1)** | **1.0000 (1)** | 0.9954 (5) | 0.9491 (6) | 0.6836 (7) | **1.0000 (1)** | * | **1.0000 (1)** |
| Magic.gamma | 0.8853 (2) | 0.8056 (5) | 0.8304 (4) | 0.7818 (6) | 0.6784 (8) | 0.7257 (7) | 0.8504 (3) | **0.8930 (1)** |
| Ionosphere | **0.9779 (1)** | 0.9704 (3) | 0.5232 (7) | 0.4238 (8) | 0.6880 (6) | 0.7116 (5) | 0.7656 (4) | 0.9746 (2) |
| Glass | **0.9830 (1)** | 0.9355 (3) | 0.8199 (6) | 0.6891 (8) | 0.7106 (7) | 0.8934 (5) | 0.8952 (4) | 0.9606 (2) |
| Breastw | 0.9875 (2) | 0.9398 (5) | **0.9947 (1)** | * | 0.9555 (4) | 0.8872 (6) | 0.7631 (7) | 0.9859 (3) |
| Yelp | 0.7283 (3) | 0.7310 (2) | 0.5676 (6) | 0.5491 (7) | 0.6691 (5) | 0.4998 (8) | **0.7440 (1)** | 0.6888 (4) |
| Imdb | 0.6809 (2) | 0.5555 (6) | 0.5502 (7) | 0.5693 (5) | 0.5042 (8) | 0.5880 (4) | **0.6810 (1)** | 0.6399 (3) |
| MNIST-C-Fog | **1.0000 (1)** | 0.9901 (6) | 0.9999 (4) | 0.9327 (7) | 0.8011 (8) | **1.0000 (1)** | **1.0000 (1)** | 0.9968 (5) |
| MNIST-C-canny-Edges | 0.9977 (2) | 0.9578 (6) | 0.9649 (5) | 0.7623 (7) | 0.6864 (8) | 0.9667 (4) | **0.9986 (1)** | 0.9846 (3) |
| MVTec-AD-Zipper | **0.9307 (1)** | 0.8626 (3) | 0.7728 (6) | 0.5132 (8) | 0.7402 (7) | 0.7815 (5) | 0.8993 (2) | 0.8455 (4) |
| MNIST-C-Stripe | **1.0000 (1)** | **1.0000 (1)** | **1.0000 (1)** | 0.9159 (8) | 0.9743 (7) | **1.0000 (1)** | **1.0000 (1)** | 0.9998 (6) |
| Http | **1.0000 (1)** | **1.0000 (1)** | 0.9981 (4) | * | 0.5595 (6) | † | 0.9980 (5) | 0.9997 (3) |
| Skin | **0.9998 (1)** | 0.9995 (2) | 0.9957 (5) | * | 0.4688 (6) | † | 0.9993 (3) | 0.9990 (4) |
| Fraud | **0.9526 (1)** | 0.9228 (5) | 0.9136 (6) | 0.7745 (7) | 0.9281 (4) | † | 0.9478 (2) | 0.9451 (3) |
| Cover | **0.9994 (1)** | 0.9930 (5) | 0.9963 (3) | 0.9102 (6) | 0.5970 (7) | † | 0.9992 (2) | 0.9935 (4) |
| Shuttle | **0.9952 (1)** | 0.9866 (3) | 0.9789 (5) | 0.9383 (6) | 0.9068 (7) | † | 0.9855 (4) | 0.9922 (2) |
| AVGRank | 1.43 | 3.90 | 4.60 | 6.57 | 6.70 | 4.90 | 3.63 | 3.33 |
| p-value | NA | 0.00029508 (+) | 0.00146843 (+) | 0.00000048 (+) | 0.00000005 (+) | 0.00021790 (+) | 0.00146843 (+) | 0.00146843 (+) |

Values marked with † indicate that no result was available within 12 hours.
Values marked with * indicate that a runtime error occurred during execution.

*Table 11.* This table presents the results of all algorithms using the default parameters outlined in the original paper and **10%** of labeled anomalies.

| Dataset | WSAD-DT | DeepSAD | DevNet | FeaWAD | GANAnomaly | PreNet | ROSAS | XGBOOST |
|---|---|---|---|---|---|---|---|---|
| Optdigits | 0.9999 (3) | 0.9948 (6) | 0.9997 (4) | 0.9983 (5) | 0.6044 (8) | **1.0000 (1)** | **1.0000 (1)** | 0.9931 (7) |
| Lymphography | **1.0000 (1)** | **1.0000 (1)** | 0.9961 (4) | 0.9787 (6) | 0.8953 (7) | **1.0000 (1)** | * | 0.9903 (5) |
| Pendigits | **0.9998 (1)** | 0.9931 (4) | 0.9889 (5) | 0.9841 (6) | 0.8323 (8) | 0.9800 (7) | 0.9993 (2) | 0.9971 (3) |
| Vertebral | **0.9051 (1)** | 0.7810 (5) | 0.4974 (7) | 0.6399 (6) | 0.3618 (8) | 0.9036 (2) | 0.8377 (3) | 0.8125 (4) |
| Wdbc | **1.0000 (1)** | 0.9990 (3) | **1.0000 (1)** | 0.9979 (4) | 0.9830 (8) | 0.9969 (6) | 0.9933 (7) | 0.9979 (4) |
| Wpbc | **0.6915 (1)** | 0.6144 (4) | 0.6403 (3) | 0.5619 (6) | 0.4635 (7) | 0.6649 (2) | * | 0.6135 (5) |
| Stamps | **0.9902 (1)** | 0.9492 (4) | 0.8909 (6) | 0.8664 (7) | 0.7529 (8) | 0.9697 (2) | 0.9544 (3) | 0.9011 (5) |
| Satimage-2 | **0.9973 (1)** | 0.9715 (5) | 0.9562 (6) | 0.9753 (4) | 0.9783 (3) | 0.9243 (8) | 0.9526 (7) | 0.9866 (2) |
| Spambase | 0.9615 (2) | 0.9073 (5) | 0.9310 (4) | * | 0.6103 (7) | 0.8781 (6) | 0.9412 (3) | **0.9686 (1)** |
| Thyroid | 0.9957 (2) | 0.9705 (5) | 0.9753 (4) | 0.9449 (6) | 0.7761 (8) | 0.9407 (7) | **0.9966 (1)** | 0.9843 (3) |
| Mnist | **0.9921 (1)** | 0.9511 (4) | 0.9425 (5) | 0.8121 (8) | 0.8284 (7) | 0.9368 (6) | 0.9703 (3) | 0.9914 (2) |
| Yeast | **0.6819 (1)** | 0.6673 (2) | 0.5805 (6) | 0.5260 (7) | 0.4885 (8) | 0.6633 (3) | 0.6593 (4) | 0.6471 (5) |
| Cardio | 0.9945 (2) | 0.9869 (3) | 0.9830 (4) | 0.7676 (8) | 0.9088 (7) | 0.9428 (6) | 0.9743 (5) | **0.9967 (1)** |
| Vowels | **0.9736 (1)** | 0.9712 (2) | 0.8839 (6) | 0.7696 (8) | 0.8231 (7) | 0.9300 (5) | 0.9653 (3) | 0.9516 (4) |
| Wine | **1.0000 (1)** | **1.0000 (1)** | 0.9954 (5) | 0.9491 (6) | 0.6836 (7) | **1.0000 (1)** | * | **1.0000 (1)** |
| Magic.gamma | 0.9187 (2) | 0.8919 (4) | 0.8323 (6) | 0.7932 (7) | 0.6680 (8) | 0.8594 (5) | 0.9141 (3) | **0.9203 (1)** |
| Ionosphere | **0.9769 (1)** | 0.9693 (2) | 0.3976 (8) | 0.6145 (7) | 0.6887 (6) | 0.8720 (4) | 0.8587 (5) | 0.9605 (3) |
| Glass | **0.9830 (1)** | 0.9355 (3) | 0.8199 (6) | 0.6891 (8) | 0.7106 (7) | 0.8934 (5) | 0.8952 (4) | 0.9606 (2) |
| Breastw | 0.9876 (3) | 0.9670 (4) | **0.9943 (1)** | * | 0.9549 (5) | 0.9388 (6) | 0.8695 (7) | 0.9925 (2) |
| Yelp | **0.8671 (1)** | 0.7777 (3) | 0.7061 (4) | 0.5846 (8) | 0.6691 (6) | 0.6491 (7) | 0.6888 (5) | 0.8351 (2) |
| Imdb | 0.8140 (2) | 0.8094 (3) | 0.6801 (5) | 0.5792 (7) | 0.5021 (8) | 0.6128 (6) | **0.8230 (1)** | 0.7616 (4) |
| MNIST-C-Fog | **1.0000 (1)** | **1.0000 (1)** | **1.0000 (1)** | 0.6645 (8) | 0.8040 (7) | **1.0000 (1)** | **1.0000 (1)** | **1.0000 (1)** |
| MNIST-C-canny-Edges | **0.9999 (1)** | 0.9946 (6) | 0.9980 (5) | 0.7528 (7) | 0.6962 (8) | 0.9995 (3) | 0.9998 (2) | 0.9988 (4) |
| MVTec-AD-Zipper | **0.9278 (1)** | 0.8923 (4) | 0.7327 (6) | 0.5681 (8) | 0.7385 (5) | 0.7313 (7) | 0.8945 (3) | 0.8964 (2) |
| MNIST-C-Stripe | **1.0000 (1)** | **1.0000 (1)** | **1.0000 (1)** | 0.8470 (8) | 0.9738 (7) | **1.0000 (1)** | **1.0000 (1)** | **1.0000 (1)** |
| Http | **1.0000 (1)** | **1.0000 (1)** | 0.9984 (4) | * | 0.5482 (6) | † | 0.9980 (5) | 0.9992 (3) |
| Cover | 0.9997 (2) | 0.9976 (4) | 0.9990 (3) | * | 0.4632 (6) | † | **0.9998 (1)** | 0.9968 (5) |
| Skin | **0.9998 (1)** | 0.9995 (3) | 0.9949 (5) | * | 0.5026 (6) | † | 0.9993 (4) | 0.9996 (2) |
| Fraud | 0.9566 (3) | 0.9337 (5) | 0.9342 (4) | 0.8545 (7) | 0.9181 (6) | † | 0.9641 (2) | **0.9680 (1)** |
| Shuttle | 0.9949 (3) | 0.9956 (2) | 0.9779 (5) | 0.9044 (7) | 0.9714 (6) | † | 0.9849 (4) | **0.9995 (1)** |
| AVGRank | 1.47 | 3.33 | 4.47 | 6.87 | 6.83 | 4.83 | 3.80 | 2.87 |
| p-value | NA | 0.00031712 (+) | 0.00028952 (+) | 0.00000005 (+) | 0.00000005 (+) | 0.00020799 (+) | 0.00175194 (+) | 0.02846843 (+) |

Values marked with † indicate that no result was available within 12 hours.
Values marked with * indicate that a runtime error occurred during execution.

## O. Parameter Sensitivity

In WSAD-DT, each of the two class centers (normal and anomalous) is associated with a light-tailed kernel that enforces tighter clustering around that center. We allow each center's light-tailed kernel to have its own bandwidth parameter, enabling the normal and anomalous clusters to have different degrees of "compactness." Both centers have the same parameter $\nu$ for the single heavy-tailed kernel to model out-of-class similarity, ensuring a broader margin is maintained for any point that does not align with its assigned center. Similarly, for the diversity term (which penalizes excessive clustering), we use class-specific bandwidth parameters that can adjust how strongly intra-class distances are regulated. We conduct a detailed ablation to analyze the effect of varying these parameters, showing that while the method is robust across a reasonable range of values, fine-tuning can further improve performance in different data regimes.

**Experimental Setup.**

- **Normal Kernel Bandwidth:** We vary the bandwidth $\sigma_{\mathcal{U}}$ of the light-tailed kernel (assigned to the normal center) within $\{0.1, 0.25, 0.5, 0.75\}$.

- **Anomaly Kernel Bandwidth:** We similarly vary $\sigma_{\mathcal{A}}$, the bandwidth of the anomalous center's light-tailed kernel, within $\{0.25, 0.5, 0.75, 1.0\}$.

- **Heavy-Tailed Kernel (Tail Parameter $\nu$):** We adjust the degrees-of-freedom parameter $\nu$ of the heavy-tailed kernel over $\{0.2, 0.5, 0.75, 1.0\}$.

- **Diversity Bandwidth:** Finally, we vary the bandwidth $(\sigma_{\text{div},\mathcal{U}}, \sigma_{\text{div},\mathcal{A}})$ used in the exponential term of the diversity loss in $\{0.1, 0.5, 1.0, 2.0\}$.

All other hyperparameters (e.g., network architecture, batch size, and ensemble splits) are kept at their default settings.

**Results and Analysis.** Figure 4 illustrates the impact of changing each parameter on AUC-ROC for five example datasets. We highlight several observations:

- **Light-tailed kernels (Normal vs. Anomaly Center).** Although performance does change with different bandwidth values, the method remains generally robust for $\sigma \in [0.2, 1.0]$. For normal data a smaller $\sigma$ often yields higher accuracy by enforcing a tighter, more coherent cluster; however, using a larger $\sigma$ can make the model overly permissive and degrade performance. By contrast, for anomalous data, a relatively larger $\sigma$ tends to improve performance by accommodating the greater heterogeneity of anomalies, whereas a too-small $\sigma$ risks overly constraining the anomalous region and thus degrading results.

- **Heavy-Tailed Kernel (Tail Parameter $\nu$).** We find that values of $\nu$ around 0.2–1.0 often offer strong performance, preserving sufficient separation for out-of-class points.

- **Diversity Term ($\sigma_{\text{diversity}}$).** For most datasets, $\sigma_{\text{diversity}} \in [0.5, 1.0]$ provides a balanced penalty that prevents collapse while still allowing appropriate clustering within each class. Extremely small values (e.g., 0.1) risk over-penalizing points that are naturally close, while very large values reduce the diversity penalty, making the model more susceptible to trivial collapse in extreme cases.

**Practical Recommendations.** Overall, WSAD-DT is not highly sensitive to modest changes in these parameters; even when they are set sub-optimally, the method continues to outperform or stay on par with strong baselines. We recommend the following default ranges, based on empirical results:

- $\sigma_{\mathcal{U}}$, $\sigma_{\mathcal{A}} \approx 0.5$ and $1.0$.

- $\nu$ (heavy-tailed) in the range $\{0.2, 1.0\}$.

- $\sigma_{\text{diversity},\mathcal{U}} \approx 0.5$ and $\sigma_{\text{diversity},\mathcal{A}} \approx 1.0$.

In practice, a quick grid or random search within these ranges often suffices to achieve robust performance without extensive tuning.

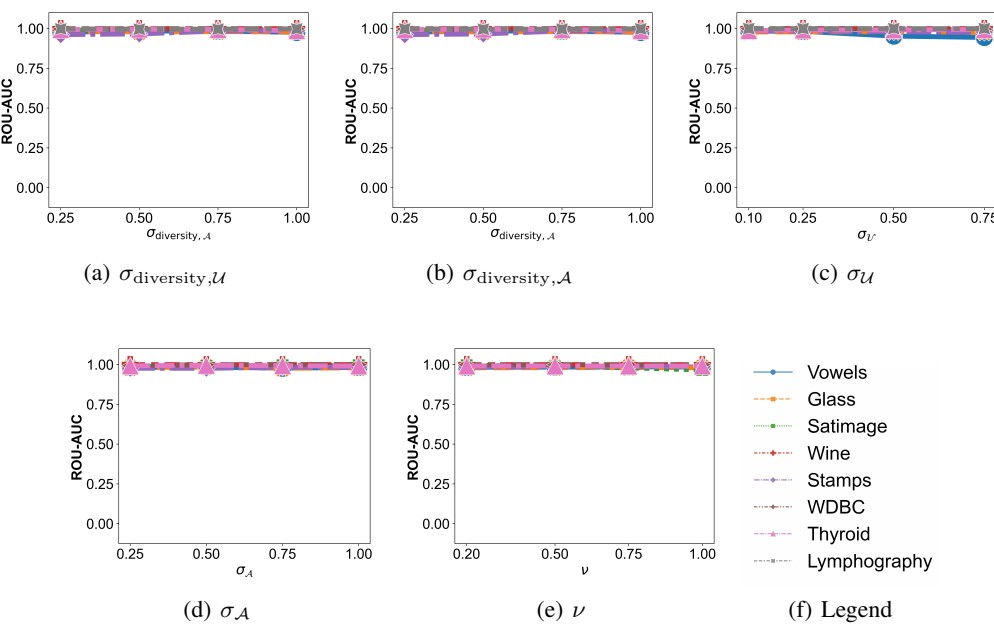

*Figure 4.* Analyzing the performance of WSAD-DT under different hyperparameters.

## P. Ablation study kernel regularization

As established in Appendix D, relying solely on the separation loss can lead to degeneracies where samples of each class collapse to a single latent point. To mitigate this, WSAD-DT includes a *kernel-based regularization term*—the diversity loss—which penalizes over-concentration and thus preserves intra-class variability. In this ablation study, we assess how omitting this diversity term affects performance.

**Experimental Setup.** We compare WSAD-DT with and without the diversity term, keeping all other hyperparameters unchanged. For both settings, we set the ensemble size of WSAD-DT to 1. Specifically, *No Regularization* removes the exponential pairwise penalty $\exp\left(-\|f_\theta(x_i) - f_\theta(x_j)\|^2/\sigma^2\right)$ from the total loss. We evaluate both variants on multiple real-world datasets under the same weakly supervised setting described in Section H

**Results.** Table 12 reports AUC-ROC scores with ("WSAD-DT") and without ("No Regularization Term") the diversity penalty. In most datasets, WSAD-DT achieves higher AUC-ROC than its unregularized counterpart. Although in a few cases, the version without regularization narrowly outperforms the regularized one, most datasets strongly benefit from including the diversity term. Specifically: Without regularization, the model can map each entire class to a single point in latent space, artificially minimizing the separation loss (Appendix D). This often degrades generalization to unseen anomalies (Goyal et al., 2020). Overall, these results confirm the theoretical and empirical rationale for using kernel-based regularization: while *some* datasets may not strictly require it to avoid degenerate solutions, it significantly increases robustness and yields better or comparable AUC-ROC on average.

*Table 12.* This table presents the AUC-ROC results of WSAD-DT with and without diversity term.

| Dataset | WSAD-DT | No regularization term |
|---|---|---|
| Optdigits | **0.9984 (1)** | 0.9864 (2) |
| Lymphography | 0.9961 (2) | **0.9981 (1)** |
| Pendigits | **0.9997 (1)** | 0.9768 (2) |
| Vertebral | 0.8827 (2) | **0.8836 (1)** |
| Wdbc | **1.0000 (1)** | 0.9928 (2) |
| Cardiotocography | **0.9347 (1)** | 0.8839 (2) |
| Wpbc | 0.6801 (2) | **0.6827 (1)** |
| Stamps | **0.9841 (1)** | 0.9765 (2) |
| Satimage-2 | **0.9954 (1)** | 0.9898 (2) |
| Spambase | **0.9317 (1)** | 0.9053 (2) |
| Thyroid | **0.9911 (1)** | 0.9732 (2) |
| Mnist | **0.9820 (1)** | 0.9249 (2) |
| Yeast | **0.6566 (1)** | 0.6413 (2) |
| Cardio | **0.9908 (1)** | 0.9514 (2) |
| Vowels | **0.9824 (1)** | 0.9670 (2) |
| Wine | **1.0000 (1)** | **1.0000 (1)** |
| Magic.gamma | **0.8974 (1)** | **0.8974 (1)** |
| Ionosphere | **0.9836 (1)** | 0.9674 (2) |
| Glass | 0.9794 (2) | **0.9919 (1)** |
| Breastw | **0.9828 (1)** | 0.9758 (2) |
| Yelp | **0.7811 (1)** | 0.6415 (2) |
| Imdb | **0.6882 (1)** | 0.6262 (2) |
| MNIST-C-Fog | **1.0000 (1)** | **1.0000 (1)** |
| MNIST-C-canny-Edges | **0.9999 (1)** | 0.9873 (2) |
| MVTec-AD-Zipper | **0.9280 (1)** | 0.8585 (2) |
| MNIST-C-Stripe | **1.0000 (1)** | **1.0000 (1)** |
| Skin | **0.9998 (1)** | 0.9992 (2) |
| Fraud | **0.9485 (1)** | 0.9064 (2) |
| Http | **1.0000 (1)** | 0.9985 (2) |
| Cover | **0.9995 (1)** | 0.9752 (2) |
| Shuttle | 0.9935 (2) | **0.9956 (1)** |
| AvgRank | 1.16 | 1.71 |
| p-value | NA | 0.000127 (+) |

## Q. Limitations

While WSAD-DT generally performs well under weak supervision, several scenarios can pose significant challenges. One issue arises when anomalies are poorly separated from normal instances, creating a fuzzy boundary that can require additional supervision or more refined feature engineering in cases of extreme overlap. Another complication occurs if anomalies are highly diverse, as we employ a single center for anomaly representation. If anomalies originate from multiple, markedly different clusters with no shared center, a single anomaly center may underfit, indicating that multi-center or multi-modal methods could be more appropriate. Finally, although WSAD-DT addresses extremely limited anomaly labels, there may be domains with such a scarcity of labeled anomalies that even the dual-tailed kernel and diversity term are not sufficient for reliable separation. Nevertheless, our experiments suggest that WSAD-DT remains robust under most practical conditions, balancing in-class compactness with broad margins and leveraging sparse anomaly labels through an ensemble mechanism.

