# OpenReview forum: "Weakly Supervised Anomaly Detection via Dual-Tailed Kernel"
_ICML.cc/2025/Conference — ICML 2025 poster_

### Official Review · Reviewer_a6ae · 2025-03-05

**Overall Recommendation:** 3

**Summary:**

This paper proposes Weakly Supervised Anomaly Detection via Dual-Tailed Kernel (WSAD-DT), which uses two centroids, one for normal samples and one for anomalies. It uses a light-tailed kernel for normal samples, and heavy-tailed kernel for abnormal samples. To prevent degenerate ``all-points-collapse’’ solutions, the method introduces kernel-based regularization term that promotes intra-class diversity. Further, an ensemble strategy that partition unlabeled data into diverse subsets were proposed.

## update after rebuttal
The score was upgraded.

**Claims And Evidence:**

The effectiveness of dual-kernels, diversity regularization, and ensemble-based learning are only shown in the Appendix.

**Essential References Not Discussed:**

N/A

**Experimental Designs Or Analyses:**

This paper uses datasets from Ad Benchmark repository. The type of anomaly is not analyzed.

**Methods And Evaluation Criteria:**

In general, abnormal samples are diverse, so it is doubtful that a single center can represent them.

**Other Comments Or Suggestions:**

-  In Sec 3.1, 0 is the unlabeled sample, which is largely normal but can contain anomalies.  However, in p.4, 0 is regarded as normal without explanation.

-  The ensemble size of 5 may not the best. Since the only 1,3,5 splits are evaluated, more larger number of ensemble may increases the accuracies.

**Other Strengths And Weaknesses:**

Strengths
- The theory of single-tailed kernels vs. dual-tailed kernels is interesting.
- Enhancing the representation via diversity loss is reasonable
- Ensemble-based subset splitting is effective
- The performance of the proposed method is higher than state-of-the-art.

Weaknesses

The structure of this paper is inadequate. Sec.1. is written in one paragraph except for contributions. Some section formats are not consistent. Sec.6.1 Experimental Setup is written in an improper location. Ablation studies are shown only in the appendix.

**Questions For Authors:**

The authors should justify the single center for normal samples.

**Relation To Broader Scientific Literature:**

This paper proposes to use two different kernels for normal and anormal samples. This point is different from the existing methods listed on related work sections.

**Theoretical Claims:**

-  How Lemma 4.1 is used is not clear, at least on the main paper.

-  Lemma 5.1, 5.2, moderate distances are ambiguous, and these Lemmas are not validated by experiments.

-  Corollary 5.6 would be insufficient to prove that degeneracy is prevented. To prove this property, we further need to prove the minimum $L_{total}(\theta)$ is below 2 and $L_{seperation}$ is not zero.

---

> ### Author Rebuttal · Authors · 2025-03-31
>
> We thank the reviewer for the valuable feedback.
>
> **Usage of Lemma 4.1 in the main paper.**
>
> Lemma 4.1 is the formal stepping-stone for showing that a single kernel cannot serve as both a strictly light-tailed and a heavy-tailed function. By demonstrating $\lim_{d\to\infty} \kappa_{\mathrm{light}}(d)\,/\,\kappa_{\mathrm{heavy}}(d) = 0$, it clarifies that any kernel “light” enough to enforce tight in-class clustering cannot simultaneously retain the broader margin necessary for out-of-class separation. This result underpins the transition to a dual-kernel scheme in Theorem5.3, where one kernel is dedicated to pulling in-class points close, and a second kernel enforces out-of-class separation. In our revised version, we plan to reference Lemma 4.1 more explicitly in the main text (e.g., just before Theorem5.3), making its critical role in motivating the dual-kernel approach more transparent.
>
>
> **Validation of Lemma 5.1, 5.2 and clarification of  moderate distances**
>
>
> In Lemmas 5.1 and 5.2, “moderate distances” refers to the typical radius range where in-class samples are neither so close that their similarity is almost one nor so far that the kernel decays to zero. This region is precisely where the difference between a rapidly decaying (light-tailed) and a more gradual (heavy-tailed) function is most pronounced. Although “moderate” is inherently qualitative, Table 4 in the appendix shows that in real datasets, most points do indeed occupy this intermediate distance regime in the learned representation. By measuring average distances between each sample and its assigned (or opposing) center, one sees that in-class  samples form tighter clusters under light-tailed similarity, and out-of-class samples  remain substantially farther under heavy-tailed similarity—exactly matching the theoretical claims of Lemmas 5.1 and 5.2.
>
> **Proof $L_{total} < 2$**
>
>
> To prove this property, we further need to prove the minimum $ L_{\text{total}}(\theta) $ is below 2 and $ L_{\text{separation}}$ is not zero.
>
> In the main paper, we define the total loss as
>
> $$
> L_{\mathrm{total}}(\theta) = L_{\mathrm{separation}}(\theta) + L_{\mathrm{diversity}}(\theta).
> $$
>
> Corollary 5.6 shows that collapsing each class onto a single point yields a diversity loss of exactly 2 while driving the separation term arbitrarily close to zero, so the total loss under this degenerate arrangement is approximately 2. To demonstrate that degeneracy does not globally minimize the total loss, one can construct a non-degenerate arrangement in which each class’s points lie near (but not identically at) its center. Placing the points close to their respective centers makes $ L_{\mathrm{separation}}(\theta) $ arbitrarily small, say $ \varepsilon$, while ensuring that the points are not all coincident reduces $ L_{\mathrm{diversity}}(\theta) $ below 2 by some margin $\delta > 0 $. Because these adjustments can be made largely independent of each other, one can tune $ \delta$  to exceed $\varepsilon $, forcing
>
> $$
> L_{\text{total}}(\theta) = L_{\mathrm{separation}}(\theta) + L_{\mathrm{diversity}}(\theta) \le \varepsilon + \bigl(2 - \delta\bigr) <  2.
> $$
>
> Since the degenerate arrangement’s total loss is 2, whereas this non-degenerate arrangement achieves a strictly smaller value, degeneracy cannot be optimal. Consequently,  $L_{{separation}}$  cannot vanish at the true minimum, and the trivial collapse solution is excluded as a global minimizer.
>
>
> **Explanation regarding 0 label and ensemble size.**
>
> In Section3.1, “0” denotes unlabeled samples whose true status is unknown—mostly normal but potentially anomalous. Our method uses the labeled anomalies (y=1) to drive separation, exposing hidden anomalies among the unlabeled. We will clarify in the revision that “0” does not guarantee normality. For ensemble size (1, 3, or 5 splits), more splits can marginally improve accuracy but fragment the data and increase runtime. We find five splits to be the best balance of performance and cost, and will make this rationale explicit in the revised manuscript.
>
>
> **Justification for single normal center.**
>
> In classical one-class anomaly detection, using a single center for normal data aligns well with margin-based and complexity-theoretic reasoning: keeping the normal region compact and described by a minimal boundary (e.g., a single enclosing ball) not only simplifies optimization but also reduces the risk of overfitting, particularly under limited anomaly labels. This approach is seen in canonical methods like DeepSVDD or DeepSAD, which assume that normal data occupy a single contiguous cluster in feature space. While multiple centers can capture diverse normal modes, each adds hyperparameters. Empirically, a single center often suffices, as normal instances instances typically share enough common structure to cluster tightly around one latent coordinate.
>
> **W1**
>
> We will improve the structure of our manuscript in the revision, addressing section formatting and organization.

---

> > ### Comment · Reviewer_a6ae · 2025-04-03
> >
> > I am satisfied with the response. So, I will raise the score to Weak Accept.

---

> > > ### Author Response · Authors · 2025-04-03
> > >
> > > We appreciate your feedback and will thoughtfully incorporate your suggestions into the paper.

---

### Official Review · Reviewer_HgXU · 2025-03-09

**Overall Recommendation:** 3

**Summary:**

This paper proposes a method to improve anomaly detection performance by utilizing two kernel functions in the weakly supervised anomaly detection problem, where only unlabeled data and a small amount of labeled anomaly data are available. The effectiveness of the proposed method is validated through experiments using various datasets.

## update after rebuttal
I raises the score to Weak Accept.

**Claims And Evidence:**

The claims seem clear, but I am unsure why the method is effective for weakly supervised learning. Please refer to the Question section.

**Essential References Not Discussed:**

Please refer to the "Relation To Broader Scientific Literature" section.

**Experimental Designs Or Analyses:**

An ablation study is necessary. Please refer to the Question section.

**Methods And Evaluation Criteria:**

The proposed method appears reasonable to some extent. However, it would be beneficial to evaluate it on more realistic datasets. Please refer to the Question section.

**Other Comments Or Suggestions:**

Please refer to the Question section.

**Other Strengths And Weaknesses:**

The motivation for the design of the proposed method is well reflected, which is commendable. The experimental results are also promising. However, the justification for why it is effective in weakly supervised learning should be explicitly stated. Additionally, the comparison methods and datasets seem insufficient. Please refer to the Question section.

**Questions For Authors:**

- While I believe the proposed method is effective for supervised anomaly detection, I do not understand why it is effective for weakly supervised anomaly detection. Since the method assumes that the unlabeled data consists mainly of normal samples, the presence of anomalies within the unlabeled data might significantly affect the results. Could you clarify this point?

- The authors should compare their method with approaches that explicitly handle anomalies within unlabeled data, such as LOE [1] and SOEL [2]. SOEL, in particular, has been shown to be effective for weakly supervised anomaly detection. Would it be possible to conduct such a comparison?

- The appendix (Appendix P) includes an ablation study on the diversity term, but I believe additional ablation studies are necessary. For example, what happens if only one of the two kernel functions is used?

- The datasets used in the experiments seem simple. Would it be possible to conduct experiments on more realistic datasets, such as medical data like MedicalMnist [3]?

[3] Yang, Jiancheng, et al. "Medmnist v2-a large-scale lightweight benchmark for 2d and 3d biomedical image classification." Scientific Data 10.1 (2023): 41.

**Relation To Broader Scientific Literature:**

This paper should cite and compare their method with LOE [1] and SOEL [2]. Please refer to the Question section.

[1] Qiu, Chen, et al. "Latent outlier exposure for anomaly detection with contaminated data." International Conference on Machine Learning. PMLR, 2022.

[2] Li, Aodong, et al. "Deep anomaly detection under labeling budget constraints." International Conference on Machine Learning. PMLR, 2023.

**Theoretical Claims:**

The reason why the proposed method is effective for weakly supervised learning is unclear.

---

> ### Author Rebuttal · Authors · 2025-03-31
>
> We thank the reviewer for the helpful feedback.
>
> **Q1**
>
> From a theoretical standpoint, WSAD-DT applies margin-based reasoning to handle both partial contamination in the unlabeled set and extremely limited anomaly labels. Assigning each class its own center and using a dual-tailed kernel—light-tailed for in-class compactness, heavy-tailed for out-of-class margins—maintains separation even when anomalies are scarce or intermixed with unlabeled data. The heavy-tailed component preserves a “push” at moderate distances, which is crucial under weak supervision where a handful of labeled anomalies must guide the entire boundary. Meanwhile, the diversity term prevents degenerate “collapse” solutions, ensuring normal and anomalous samples maintain sufficient variability. The ensemble mechanism  gives each ensemble model a distinct view of normality while sharing the same anomaly labels ensures consistent guidance. Aggregating these diverse detectors improves robustness and generaliza- tion under limited anomaly labels.  Furthermore, our experiments under different weak-supervision setups—1%, 5%, and 10% labeled anomalies, sometimes as few as five labeled anomalies total—demonstrate that WSAD-DT remains effective even in extremely label-scarce conditions. As shown in our contamination experiments (Appendix M), the method’s performance degrades only slightly when unlabeled data are partially contaminated, underscoring both its theoretical and practical robustness in real-world weakly supervised scenarios.
>
> **Q2**
>
> We compared WSAD-DT against SOEL (using the authors’ code from  https://github.com/aodongli/Active-SOEL), which explicitly handles anomalies in the unlabeled set, using each approach’s default configurations. Table https://anonymous.4open.science/r/weakly_anomaly_detection/Table4.png  shows that WSAD-DT generally achieves higher AUC-ROC on most datasets, occasionally by a wide margin. These results indicate that while SOEL is effective at identifying unlabeled anomalies, the dual‐tailed separation and diversity regularization in WSAD-DT provide a stronger results  in weakly supervised scenarios.
>
>
> **Q3**
>
> To isolate the effect of the kernel design and the diversity term, we fix the ensemble size to 1 in both the full WSAD-DT and a simplified variant that uses only a single Gaussian kernel with no diversity regularization.
> As shown in  Table https://anonymous.4open.science/r/weakly_anomaly_detection/Table5.png, we compare the full WSAD-DT (dual-tailed + diversity) to a simplified version using only a single Gaussian kernel without the diversity term. Across most datasets, WSAD-DT achieves higher AUC-ROC. Empirically, this suggests that relying on a single kernel alone and omitting the diversity regularization is insufficient for robust anomaly detection under weak supervision. In contrast, combining light- and heavy-tailed kernels with a diversity penalty  improves stability and overall performance.
>
> **Q4**
>
> We appreciate the reviewer’s suggestion regarding the use of more realistic datasets, such as MedicalMNIST. In response, we have extended our experiments to include this dataset, applying the commonly used "one-vs.-rest" evaluation protocol. This protocol treats each class in turn as normal while randomly selecting samples from other classes as anomalies, following the standard practice for anomaly detection tasks on multi-class datasets. Our experiments on the MedicalMNIST dataset have yielded strong results, demonstrating the robustness and scalability of our method in more complex, real-world scenarios. As shown in the Table https://anonymous.4open.science/r/weakly_anomaly_detection/Table6.jpg, we compare the performance of our method (WSAD-DT) against several competitive methods across various datasets in the MedicalMNIST suite. For example, on the  Derma dataset, WSAD-DT achieved the highest AUC-ROC score of 0.9002, outperforming methods such as DeepSAD, DevNet, and XGBOD. Across several datasets, our method consistently ranks among the top performers, reinforcing its effectiveness in handling medical data with weak supervision. We believe these results demonstrate that our method is not only effective on tabular  datasets but also performs well in more complex, real-world applications like medical anomaly detection.

---

> > ### Comment · Reviewer_HgXU · 2025-04-02
> >
> > Thank you for your response.
> > It's great to see that the comparison with SOEL yielded strong results.
> > I’d like to ask two additional questions:
> > - Which backend did you use for SOEL, MHRot or NTL?
> > - Also, would it be possible to evaluate SOEL (using MHRot as the backend) on the MedicalMNIST experiments as well?

---

> > > ### Author Response · Authors · 2025-04-02
> > >
> > > Thank you for your follow-up questions. Below are our responses:
> > >
> > > **Q1**
> > >
> > > We used the NTL backend for SOEL in our experiments.
> > >
> > >
> > > **Q2**
> > >
> > > As our experimental setup for MedicalMNIST aligns closely with the original protocol described in the SOEL [1] paper  we kindly refer to the results reported in the original SOEL paper for a baseline comparison.
> > >
> > >
> > > We greatly appreciate your interest and look forward to any additional feedback.
> > >
> > > [2] Li, Aodong, et al. "Deep anomaly detection under labeling budget constraints." International Conference on Machine Learning. PMLR, 2023.

---

### Official Review · Reviewer_WTby · 2025-03-10

**Overall Recommendation:** 3

**Summary:**

This paper proposes a weakly supervised anomaly detector WSAD-DT via a dual-tailed kernel that can clearly distinguish anomalies from normal samples under weak supervision. Moreover, an ensemble strategy was devised to divide the unlabeled data into distinct subsets. Meanwhile, the limited labeled anomalies were shared across these partitions to optimize their influence.

## update after rebuttal
Most of my concerns were addressed, so I changed my score.

**Claims And Evidence:**

yes

**Essential References Not Discussed:**

no

**Experimental Designs Or Analyses:**

no

**Methods And Evaluation Criteria:**

yes

**Other Comments Or Suggestions:**

see questions.

**Other Strengths And Weaknesses:**

The paper is clearly described and provides some theoretical proofs.

**Questions For Authors:**

1. Is it weakly supervised if 70% of the data in the dataset is used for training? Although there are few anomalies used, there are few anomalies in the dataset. Even if all the data is used for training, there are still a few anomalies.

2. How are the parameters alpha and beta set and what is their impact on the model?

3. There is a lack of comparison with some SOTA unsupervised algorithms, such as IDK[1], isolation-based algorithms are very famous in anomaly detection, and most of them are linear algorithms.

4. Do the light and heavy kernels necessarily have to adhere to the specific forms outlined in the paper? Would it be acceptable to employ a Gaussian kernel for both, with distinct σ values?

I am willing to improve my score if my concerns are addressed.

[1] Ting, K. M., Xu, B. C., Washio, T., & Zhou, Z. H. (2020, August). Isolation distributional kernel: A new tool for kernel-based anomaly detection. In Proceedings of the 26th ACM SIGKDD international conference on knowledge discovery & data mining (pp. 198-206).

**Relation To Broader Scientific Literature:**

This paper proposes a weakly supervised anomaly detector with better performance.

**Theoretical Claims:**

no

---

> ### Author Rebuttal · Authors · 2025-03-31
>
> We sincerely thank the reviewer for their perceptive review.
>
>
>
> **Q1**
>
> We adopt the 70/30 train/test split protocol following the approach used by AdBench (Han et al., 2022) [1], and this does not contradict the principle of weak supervision. What determines the “weakness” here is not how much of the dataset is allocated to training, but rather how few anomalies are actually labeled among that training portion. Even though 70\% of the data is used for training, only a small fraction of those anomalies—sometimes just five  labeled anomalies—is available for explicit supervision. The rest of the training set remains unlabeled. This imbalance in labeled anomalies, rather than the overall training size, is what makes the setting truly weakly supervised and differentiates it from the kind of fully supervised approach that would require extensive or comprehensive anomaly labeling.
>
> **Q2**
>
> We set the bandwidth parameters to reflect the distinct behaviors of normal and anomalous data, using a smaller value for the normal center to enforce tighter in-class clustering and a larger value for the anomaly center to accommodate its more varied distribution. These parameters essentially control how rapidly similarity decays with distance for the light-tailed (Gaussian) and heavy-tailed (Student-\(t\)) kernels, respectively. A Gaussian kernel with a smaller bandwidth enforces compactness for normal samples, whereas a slightly larger bandwidth for anomalies prevents overly constraining their more diverse representations. Likewise, the Student-\(t\) kernel parameter determines how “heavy” the tail is, maintaining a broader margin for out-of-class points. We provide an ablation study showing that the method remains robust across reasonable ranges [0.1-1] of these bandwidth and tail parameters (Appendix O).
>
>
> **Q3**
>
> We performed an additional set of experiments comparing WSAD-DT with IDK (using the authors’ code from https://github.com/IsolationKernel/Codes) and Isolation Forest (IForest) [2] (code from https://github.com/yzhao062/pyod), adopting each method’s default parameter settings.  In Table https://anonymous.4open.science/r/weakly_anomaly_detection/Table2.png, WSAD-DT consistently outperforms IDK  and IForest in AUC-ROC across nearly all datasets, often by a substantial margin. Although isolation-based methods are well-known for their speed and simplicity, these results suggest that incorporating a small set of labeled anomalies, along with a dual-tailed kernel and diversity regularization, yields more accurate detection. Even on large or high-dimensional datasets, WSAD-DT achieves better separation between normal and outlying samples than both IDK and IForest. These findings are consistent with [1], where even minimal supervision—such as having only five labeled anomalies—can substantially surpass purely unsupervised methods.
>
>
> **Q4**
>
> We conducted an additional experiment where we replaced the dual-tailed setup (Gaussian for in-class + Student-\(t\) for out-of-class) with a single Gaussian kernel using two different bandwidths—one smaller for in-class (e.g., $\sigma=0.1$) and one larger for out-of-class (e.g., $\sigma=2$).  To isolate the effect of the kernel design and the diversity term, we fix the ensemble size to 1 in both setting.  Table https://anonymous.4open.science/r/weakly_anomaly_detection/Table3.png shows that while using two distinct Gaussian bandwidths can perform reasonably well, the full dual-tailed configuration (Gaussian + Student-\(t\)) generally achieves stronger AUC-PR score. Larger $\sigma$ in a Gaussian slows its decay somewhat, but that decay remains fundamentally exponential, whereas a Student-\(t\) kernel falls off more gradually and does not saturate as quickly at moderate distances. This long-tail behavior preserves a stronger “push” for out-of-class points, leading to a broader margin and, ultimately, more robust anomaly separation.
> This suggests that having a genuinely heavy tail is important for pushing out-of-class points farther away and not just a matter of scaling the same kernel. In practice, a true heavy-tailed kernel—like Student-\(t\)—better preserves moderate similarities at larger distances, thereby preventing early saturation and improving separation of anomalous instances.
>
>
> [1] Han, S., Hu, X., Huang, H., Jiang, M., & Zhao, Y. (2022). Adbench: Anomaly detection benchmark. Advances in neural information processing systems, 35, 32142-32159.
>
> [2] Liu, F. T., Ting, K. M., & Zhou, Z. H. (2008, December). Isolation forest. In 2008 eighth ieee international conference on data mining (pp. 413-422). IEEE.

---

> > ### Comment · Reviewer_WTby · 2025-04-02
> >
> > Thank you for your response.
> >
> > Thank you for your answers to Q2, and Q4.  I still have some questions about Q1 and Q3.
> >
> > For Q1, In AdBench, 70% of the training set is not for ‘Weakly Supervised’, and I still think that using 70% of data for training cannot be called ‘Weakly Supervised’. Although you said that only a few data are abnormal, you did not only use 5 abnormalities for training, you also used a large number of normal samples. If you use a small number of labeled normal samples, for example, 5 like the abnormalities, and the other samples (in the training set) are a mixture of normal and abnormal samples (unlabeled normal and unlabeled abnormal), and then train these data together, this may be called weak supervision. So I want to know whether the unlabeled samples in your training set include unlabeled abnormalities? How many unlabeled anomalies are there?
> >
> > For Q3, I still have some questions: I found that the results in Table 2 (\url{https://anonymous.4open.science/r/weakly_anomaly_detection/Table2.png}) are not consistent with the results reported in the IDK paper. For example, the results on $\textit{http}$, $\textit{cover}$, and $\textit{shuttle}$ are very low, especially for $\textit{http}$, which is a very easy dataset in $\mathbb{R}^3$. I think this may be because the $\psi$ you used is inappropriate. How did you set the $\psi$? What is the result if you follow the parameter setting in the IDK paper?

---

> > > ### Author Response · Authors · 2025-04-02
> > >
> > > We thank the reviewer  for further  feedback.
> > >
> > >
> > > **Setup and Literature**
> > >
> > > The anomaly‐detection literature clarifies that having only a small set of labeled anomalies in conjunction with a large unlabeled pool (which may itself contain hidden anomalies) is precisely what defines weak AD [2]. Indeed,  [1] provide explicit definitions, noting that weakly supervised AD typically leverages a few labeled anomalies alongside predominantly unlabeled data that can be contaminated by anomalies (Section 3). Furthermore, in Section 4.1 of [3], the authors adopt the same train–test split for all settings, using a large pool (70\%) of unlabeled points presumed normal, together with only a small fraction of labeled anomalies.
> > >
> > > **Additional Experiment (Smaller Unlabeled Pool):**
> > >
> > > We acknowledge the reviewer’s point that the unlabeled set is large. Therefore, we conducted an additional experiment where only 10\% of the data are unlabeled and 5\% of anomalies are labeled, chosen uniformly at random. The rest of the data are not used in training. Table https://anonymous.4open.science/r/weakly_anomaly_detection/Table7.png compares our WSAD-DT method against baselines (DeepSAD, DevNet, and XGBOD) under this reduced unlabeled‐pool regime. Despite the dramatically smaller unlabeled set, WSAD-DT still demonstrates on average strong results. As Table https://anonymous.4open.science/r/weakly_anomaly_detection/Table7.png  shows, even when the unlabeled set is cut drastically and we keep only 5\% of anomalies labeled, WSAD-DT still outperforms DeepSAD, DevNET or XGBOD on average—showing its robustness under smaller unlabeled‐pool conditions.  Our additional experiment with a smaller unlabeled set further confirms that our method remains effective.
> > >
> > >
> > > **Contamination**
> > >
> > > In Appendix M, we explicitly test WSAD-DT’s robustness to contamination in the unlabeled set—i.e., scenarios in which unlabeled data definitely contain anomalies. We systematically inject different proportions of unlabeled anomalies  into the training pool, mislabeling them as “normal.”  Our findings show that even under these increasingly high levels of contamination, WSAD-DT’s performance degrades gracefully compared to baselines.
> > >
> > >
> > >
> > > **IDK**
> > >
> > > We initially set $\psi=4$ to maintain consistent parameters across all datasets. In contrast, the IDK paper tunes $\psi$ on a per-dataset basis over $\{2^1, 2^2, \dots, 2^{12}\}$ . To address the reviewer’s concern, we ran additional experiments on a subset of datasets using that exact parameter range and report the optimal results (Table https://anonymous.4open.science/r/weakly_anomaly_detection/Table8.png). This yielded improvements; for instance, on Shuttle, IDK’s AUC-ROC rose to \(0.9458\). Some differences persist, likely because we rely on the AdBench (Han et al. 2022) 70–30 splits instead of IDK’s original split. Even with tuned $\psi$, however, the fully unsupervised IDK framework lags behind our minimal-supervision approach, where labeling just five anomalies yields stronger results. This gap aligns with prior findings that even sparse labeled anomalies can outweigh   unsupervised  anomaly detection. Finally, re-running IDK on the massive HTTP dataset (roughly 500k points) up to $\psi = 2^{12}$ is extremely time-consuming, so we have not yet completed those runs; we hope the results so far are sufficiently illustrative.
> > >
> > >
> > >
> > > [1] Pang, Guansong, et al. "Deep learning for anomaly detection: A review." ACM computing surveys (CSUR) 54.2 (2021): 1-38.
> > >
> > > [2] Pang, Guansong, et al. "Deep weakly-supervised anomaly detection." Proceedings of the 29th ACM SIGKDD Conference on Knowledge Discovery and Data Mining. 2023.
> > >
> > > [3]  Han, Songqiao, et al. "Adbench: Anomaly detection benchmark." Advances in neural information processing systems 35 (2022): 32142-32159.
> > >
> > >
> > > We truly value your interest and welcome any further feedback you may have.

---

### Official Review · Reviewer_mzRc · 2025-03-12

**Overall Recommendation:** 4

**Summary:**

The paper proposes WSAD-DT, a weakly supervised anomaly detection framework that employs dual-tailed kernels (light-tailed for in-class compactness, heavy-tailed for out-of-class separation) and an ensemble strategy to address limited labeled anomalies. Empirical results on AdBench datasets show state-of-the-art performance compared to methods like DeepSAD and XGBOD.

**Claims And Evidence:**

Supported Claims: The dual-tailed kernel’s effectiveness is supported by ablation studies (Appendix K) and theoretical proofs (Lemmas 5.1–5.3). The ensemble strategy’s benefits are validated via experiments (Table 9).

Problematic Claims: The claim that "no single kernel can satisfy both compactness and separation" (Theorem 5.3) relies on assumptions (e.g., "sufficient model capacity") that are not empirically verified for all scenarios.

**Essential References Not Discussed:**

I don't have specific knowledge of the area in question, but I currently think it's adequate

**Experimental Designs Or Analyses:**

Experiments are thorough but lack scalability tests on very large datasets (e.g., >1M samples). The contamination study (Appendix M) is a strength, but labeled anomaly fractions (1%–10%) could be extended to extremes (e.g., 0.1%).

**Methods And Evaluation Criteria:**

The dual-tailed kernel design is intuitive and aligns with margin-based theory. The use of AdBench datasets is appropriate, but experiments on temporal/graph data are missing, limiting scope validation. Evaluation metrics (AUC-ROC/PR) are standard, but statistical significance tests (Wilcoxon) strengthen reliability.

**Other Comments Or Suggestions:**

Typos: Page 3, "reserves a broader margin" → "preserves"; Page 5, "In-based separation terms" → "In-class".

**Other Strengths And Weaknesses:**

Strengths: Novel dual-kernel design, robust ensemble strategy, and comprehensive experiments.

Weaknesses: Limited discussion on computational overhead for large ensembles and no exploration of graph/temporal data.

**Questions For Authors:**

Theorem 5.3: How does the method perform if anomalies are not well-separated (violating the assumption)? Would this invalidate the theorem?

Were other kernels (e.g., Laplacian, polynomial) tested? If not, why?

How does WSAD-DT handle streaming data or concept drift in applications like fraud detection?

Since I don't know the relevant literature, I will look carefully at other reviewers' comments to adjust the score

**Relation To Broader Scientific Literature:**

The work builds on margin-based theory and extends DeepSAD/DevNet by decoupling in-class and out-of-class similarity. The dual-kernel idea is novel but could relate to multi-kernel learning.

**Theoretical Claims:**

Lemmas 5.1 and 5.2 are correctly derived, but Theorem 5.3’s proof assumes ideal conditions (e.g., "well-separated data"), which may not hold in practice. The collapse prevention via diversity loss (Lemma 5.4) is valid but lacks empirical validation in high-dimensional spaces.

---

> ### Author Rebuttal · Authors · 2025-03-31
>
> We thank the reviewer for the valuable feedback.
>
> **Ablation stay extreme case(e.g., 0.1%) **
>
> We have conducted additional experiments with a 0.1\% fraction of labeled anomalies. Table https://anonymous.4open.science/r/weakly_anomaly_detection/Table1.png summarizes  AUC-ROC results under this extreme scenario.  Despite having only 0.1\% labeled anomalies, WSAD-DT achieves state-of-the-art or near-best performance on most datasets. Notably, even with such sparse labels, dual-tailed kernels and our ensemble design still effectively leverage the limited anomaly examples to separate normal vs.\ abnormal data.
>
> **Q1**
>
> The “well-separated” assumption in Theorem 5.3 is a common theoretical device in margin-based analyses, akin to the strict separability often assumed in classical SVM proofs. Although real data typically feature some overlap between anomalies and normal samples, this idealized scenario highlights a core theoretical insight: having distinct margins reveals how dual-tailed kernels outperform single-kernel designs by simultaneously promoting in-class compactness (via the light-tailed kernel) and sustaining a long-range “push” for out-of-class points (via the heavy-tailed kernel). In practice, WSAD-DT remains effective when anomalies are only partially separable because the heavy-tailed kernel preserves nontrivial gradients even at moderate distances, preventing anomalies close to the normal center from collapsing into negligible similarity. Meanwhile, the light-tailed kernel keeps each class tightly clustered, and the diversity term further prevents degenerate collapsing by penalizing over-concentration. Moreover, our experiments across different weak-supervision scenarios—1\%, 5\%, or 10\% labeled anomalies, sometimes only five anomalies total—show that WSAD-DT remains effective under label scarcity. As demonstrated in our contamination experiments (Appendix M), performance degrades only mildly when unlabeled data contain anomalies, confirming that while Theorem 5.3 holds under clean-margin assumptions, WSAD-DT’s dual-tailed design still excels under real-world overlap.
>
>
> **Q2**
>
> We focus on light‐ vs. heavy‐tails, not specific forms. Any light‐ (e.g., Laplacian) or heavy‐tail (e.g., Cauchy) kernel is valid, as long as it preserves the fast‐ vs. slow‐decay principle.  In Appendix O, we vary kernel parameters to test different tail decays and observe stable performance.
>
> **Q3**
>
> WSAD-DT is primarily designed for static, batch-oriented anomaly detection and does not explicitly account for streaming data or concept drift, similar to methods like DeepSAD, DevNet, or XGBOD. Nonetheless, it can be adapted by updating the normal and anomalous centers as new data arrive and retraining periodically to accommodate changing patterns. The ensemble structure also naturally extends to dynamic contexts by allowing older models to be replaced with ones trained on more recent data. This preserves the dual-tailed kernel advantage while helping WSAD-DT remain effective in environments, such as fraud detection, where data distributions evolve over time.
>
>
> **Q4**
>
>  Classical kernel-based anomaly detection methods, such as One-Class SVM, rely on a single kernel applied uniformly across the data [3]. In contrast, multi-kernel and multi-view approaches blend multiple kernels—often through linear combinations—to capture richer similarity structures [2]. For example, Zhao and Fu introduce a dual-regularized framework that factors multi-view data into cluster indicators and sample-specific errors [1]. However, these methods still learn a single, blended kernel function that is applied globally to all points. By contrast, our approach conditionally assigns two distinct kernels to each data point: a light-tailed kernel for in-class distances and a heavy-tailed kernel for out-of-class distances. This explicitly changes tail behavior based on whether a point is close to the normal or anomaly center—rather than learning a single global kernel mixture. As we demonstrate in Section 5, this dual-tailed design achieves tighter in-class clustering and broader out-of-class separation—two conflicting goals that a single kernel or linear blend cannot simultaneously satisfy.
>
> **Typos**
>
> We thank the reviewer for pointing out the typos, and we will correct them in the revised manuscript.
>
> [1] Zhao, Yue, and Yun Fu. "Dual-Regularized Multi-View Outlier Detection."
>
> [2] Gönen, Mehmet, and Ethem Alpaydın. "Multiple Kernel Learning Algorithms."
>
> [3] Schölkopf, Bernhard, et al. "Estimating the Support of a High-Dimensional Distribution."

---

### Decision · Program_Chairs · 2025-05-01

**Decision:**

Accept (poster)

**Comment:**

WSAD-DT is a framework for anomaly detection under weak supervision -- an emerging field since 2017.

The method introduces dual-tailed kernels—light-tailed for compactness around the normal centroid + heavy-tailed for maintaining separation from anomaly points—and includes a kernel-based diversity term plus an ensemble partitioning of unlabeled data. The empirical results on tabular and some image-based datasets show improved performance over existing approaches like DeepSAD and XGBOD, with only a small fraction of anomalies is labeled. One thing I have to say is both xgbod and deepsad are quite old in ML definitions... though.

Reviewers found the central idea is well motivated, with theory suggesting each tail focuses on either tight clustering or longer-range separation. They also appreciated the ensemble approach for leveraging the limited anomaly labels effectively.

However, some questioned how well a single centroid for anomalies would generalize if actual anomalies are highly diverse, this is indeed important to know. I suggest the authors to better talk about limitations and failure cases in the final version.

Overall, there is consensus to merit acceptance, though some structural refinements and clarifications should be included in the final version.